# Deep learning-based no-reference image quality assessment framework for *Cryptosporidium spp.* and *Giardia spp*

**Muhammad Amirul Aiman Asri[1], Heshalini Rajagopal[2], Norrima Mokhtar**![orcid][1]*,
**Wan Amirul Wan Mohd Mahiyiddin[1], Yvonne Ai Lian Lim[3], Masahiro Iwahashi[4],
Ryosuke Harakawa[4], Fatimah Ibrahim**![orcid][5]**, Takao Ito[6]**

**1** Department of Electrical Engineering, Faculty of Engineering, Universiti Malaya, Lembah Pantai, Kuala Lumpur, Malaysia, **2** Department of Electrical and Electronics Engineering, Mila University, Negeri Sembilan, Malaysia, **3** Department of Parasitology, Faculty of Medicine, Universiti Malaya, Lembah Pantai, Kuala Lumpur, Malaysia, **4** Department of Electrical, Electronics and Information Engineering, Nagaoka University of Technology, Nagaoka, Japan, **5** Department of Biomedical Engineering, Faculty of Engineering, Universiti Malaya, Lembah Pantai, Kuala Lumpur, Malaysia, **6** Graduate School of Advanced Science and Engineering, Hiroshima University, Higashihiroshima, Japan

* norrimamokhtar@um.edu.my

## Abstract

Image Quality Assessment (IQA) plays a critical role in image-based decision-making systems, especially in domains requiring high diagnostic precision. Effective feature information is a prerequisite for the high performance of machine learning methods in parasitic organism detection, and the quality of this feature information is influenced by the quality of the images. However, No-Reference IQA (NR-IQA) models have ignored microscopy-based datasets, particularly those involving parasitic organisms such as *Cryptosporidium spp.* and *Giardia spp.*, which are vital for public health inspection. In this study, PRIQA (Parasite ResNet-101 IQA), a novel deep learning-based NR-IQA model specifically trained on a small parasite image dataset was presented. Using Mean Opinion Scores (MOS) from twenty human evaluators, nine Deep Convolutional Neural Network (DCNN) architectures were benchmarked and identified ResNet-101 as the most robust feature extractor. The features were mapped to MOS using regression models and compared with ten state-of-the-art NR-IQA algorithms. Experimental results demonstrated that PRIQA consistently outperforms existing methods, indicating its suitability as a practical quality control tool for identifying unreliable or low-quality parasite microscopy images and supporting more consistent downstream detection and diagnostic workflows in automated inspection systems.

**Data availability statement:** The reference dataset and source code, including scripts for feature extraction, distortion generation, and regression-based quality assessment developed during this study, are publicly available in the GitHub repository at https://github.com/Amirul-777/PRIQA---Parasite-ResNet-101-Image-Quality-Assessment-Study.

**Funding:** This work was partially supported by Japan Society for the Promotion of Science (JSPS) KAKENHI Grant Number JP25K21814. The funders were involved in funding acquisition, supervision, and contributions to the writing (original draft, review, and editing).

**Competing interests:** The authors have declared that no competing interests exist.

## Introduction

Image Quality Assessment (IQA) is essential across various fields for inspection purposes, including medical imaging [1], haze detection [2], and agricultural product evaluation [3]. Xu et al. [4] highlighted that the performance of parasite detection models heavily depends on effective feature extraction, which inherently linked to the quality of input images. High-quality microscopic images with good resolution, contrast, and minimal noise provide detailed and accurate information essential for feature extraction. Conversely, low-quality images may obscure critical details, directly impacting the ability to discern fine features necessary for identifying specific parasites. This is particularly critical for inspecting parasite images, such as Cryptosporidium and Giardia in drinking water treatment plants. These parasites are major non-viral infectious agents causing parasitic diarrhea [5], often found in natural water contaminated by agricultural and animal wastes [6]. They play a significant role in disease outbreaks and are therefore a major concern for public health policies and the operations of water treatment plants [7]. Thus, accurately assessing image quality is crucial to avoid inspection errors, and automating this process with advanced models would be highly beneficial.

Recent research on microscopic images has led to models that quantify image distortion and improve image quality, enhancing pathological image assessments [8]. For example, deep convolutional neural networks (CNNs) have advanced the recognition of formed elements in microscopic images through autofocus processes using blind IQA methods [9]. A similar approach has developed to automatically quantify focus for image correction in microscopic hyperspectral images of cancer cells [10]. In the realm of parasitic images, algorithms have been created for real-time detection of microscopic parasites like Protozoa [11]. However, there is a lack of specific studies or models for IQA of parasite images, particularly Cryptosporidium spp. and Giardia spp. Parasite images are subjected to distortions due to lens distortions, such as radial and tangential distortions, which can cause straight lines to appear curved and images to appear skewed. Additionally, lens aberrations can introduce further distortions that compromise image accuracy, while increased magnification can exacerbate these issues. Grating distortions also contribute, with minimal distortion in the central region and larger distortions near the edges, consistent with geometric models of lens distortion [12].

Given the advancements in IQA for microscopic images, our study aims to fill this gap and potentially stimulate further research interest in parasitic image quality assessment, improving inspection accuracy in public health applications. Techniques for parasite image quality control able to develop using IQA algorithms, which analyze image signals to quantify visual distortions [13].

IQA methods are categorized into subjective and objective types. Subjective IQA methods, though highly accurate and dependable, are labor-intensive and time-consuming, limiting their suitability for parasite image quality control tasks. In contrast, objective IQA algorithms automatically evaluate image quality using trained models and do not require human intervention. These algorithms further classified into Full-Reference (FR-IQA), Reduced-Reference (RR-IQA), and No-Reference

(NR-IQA) categories. FR-IQA compares the entire image to a perfect or pristine reference image, RR-IQA uses partial information from a reference image, and NR-IQA assesses image quality without any reference image.

In this study, we emphasize NR-IQA, which is particularly suitable for parasitic microscopy images, as it does not rely on the availability of pristine reference images. Instead, NR-IQA predicts image quality solely based on machine learning algorithms and statistical models by analyzing intrinsic image features such as brightness, contrast, and sharpness. This makes NR-IQA especially appropriate for parasitic image analysis, where high-quality reference images are typically unavailable.

Studies have been recently done on NR-IQA for various image types such as natural image [14], underwater image [15], MRIs [16], PET scans [17] and wood images [18]. A recent novel NR-IQA method called Local Feature Descriptors-IQA, designed for both authentic and artificial distortions. This method processed images by converting them to Y, Cb, and Cr color channels, applying Human Visual System (HVS) inspired filters, and extracting local features using descriptors like speeded up robust features (SURF) and features from accelerated segment test (FAST). A regression model trained on these extracted features predicts perceptual quality scores [19]. Furthermore, a self-supervised NR-IQA method named ARNIQA (leArning distoRtion maNifold for Image Quality Assessment) designed for natural images with both synthetic and authentic distortions. ARNIQA used a novel image degradation model, and a training strategy based on the SimCLR (Simple Framework for Contrastive Learning of Visual Representations) framework. It employs a pre-trained ResNet-50 encoder and a 2-layer MLP projector to generate image representations [20].

Another NR-IQA method, IL-NIQE (Integrated Local Natural Image Quality Evaluator) used a collection of pristine naturalistic images to learn a multivariate Gaussian (MVG) model. It extracts five types of Natural Scene Statistics (NSS) features from image patches, fitting each to a local MVG model to predict local quality scores [21]. These studies have demonstrated that their proposed metrics outperformed the state-of-the-art NR-IQA methods such as Blind/Referenceless Image Spatial Quality Evaluator (BRISQUE), Blind Image Quality Index (BIQI), Blind Image Integrity Notator using DCT Statistics II (BLIINDS-II), Natural Image Quality Evaluator (NIQE), Deep Bilinear Convolutional Neural Network (DB-CNN), Ensemble of Neural Image Quality Assessment (ENIQA), No-Reference Based Image Quality Assessment (NBIQA) and Spatial-Spectral Entropy-based Quality (SSEQ). Additionally, they have shown that NR-IQA algorithms must be specifically trained for the types of images they will assess.

In specialized imaging domains, studies have adopted a similar strategy of learning a regression from image features to subjective quality scores on modestly sized, domain-specific datasets. For example, a study has been done by extracting quantitative features from CT pulmonary angiography scans and trained a Random Forest regression model to predict radiologists' MOS scores for image quality on approximately 150 cases [22]. In the general NR-IQA setting, a study by Hu et al. used Swin Transformer features with a regression head to predict continuous DMOS/MOS on standard IQA databases, explicitly describing CSIQ (866 distorted images from 30 reference images) as a small laboratory synthesized dataset [23].

In parallel, CNN-based IQA frameworks have been increasingly investigated, particularly within clinical imaging contexts. One study introduced a CNN-based IQA framework for clinical skin images with human-labeled ratings to guide perceptual quality scoring [24]. A BIQA model tailored for pathological microscopy using expert MOS presented for screen and immersion settings [8]. Oh R et al. proposed a CNN based IQA for ultra-widefield fundus (UWF) images. The model predicted the IQA scores with EfficientNet-B3 as the backbone model [25]. A comprehensive review of Medical Image Quality Assessment (MIQA) approaches emphasized the need for robust NR-IQA solutions in clinical workflows [26]. These works affirm the importance of domain-specific NR-IQA but reveal a clear gap in parasite image quality prediction, particularly for Cryptosporidium and Giardia under various distortions, which this study aims to address.

From the perspective of pathological images, CNN-based models have been widely employed for classification tasks. For instance, a CNN-based hybrid model for malaria parasite classification proposed without addressing image quality [27]. In [28], a CNN based model, namely Batch Normalization, Layer Normalization, GELU – Gaussian Error Linear

Unit – and Swish functions-based network (BLGSNet) developed to detect parasites. Another study has examined the efficiency of three CNN models, ConvNeXt Tiny, EfficientNet V2 S, and MobileNet V3 S in classifying Ascaris and Taenia, which causes Helminth infections in human body [29]. These studies demonstrate the effectiveness of CNN models in parasite image classification, thereby contributing to improved diagnostic accuracy. However, despite their success in classification, CNN models did not explore for IQA of parasite images, leaving a critical gap in ensuring the reliability of visual data used for diagnosis. Hence, this highlights the need to develop a NR-IQA method to assess parasite images. To the best of our knowledge, this is the first dedicated NR-IQA framework developed specifically for parasitic microscopy images targeting *Cryptosporidium* and *Giardia*, a critical yet underrepresented area in public health diagnostics. By tailoring NR-IQA to this domain, we aim to support more robust downstream tasks such as automated parasite detection and digital screening in resource-constrained settings.

In this paper, a NR-IQA metric tailored specifically for microscopic parasite images is proposed. A controlled dataset was constructed using reference images of *Cryptosporidium* spp. and *Giardia* spp., in which four distortion types, namely Gaussian white noise (GWN), salt and pepper noise, speckle noise, and JPEG compression, which are commonly encountered in microscopic parasite imaging due to sensor noise [8], acquisition conditions [9], and compression artifacts, were synthetically applied at nine degradation levels, resulting in a total of 1,058 images. MOS were collected from twenty human observers through subjective evaluation. Features were extracted from all images using multiple convolutional neural network architectures, including ResNet variants, Inception-ResNet-v2, GoogLeNet, AlexNet, EfficientNet-B0, and DarkNet. These features were subsequently mapped to MOS using various regression models, including linear regression, support vector machines (SVMs), and decision tree regressors. To assess the reliability of the collected MOS values, correlations with established FR-IQA metrics were examined. To ensure robustness and reliability, the proposed framework was further analyzed using statistical significance testing with Wilcoxon signed-rank and Nemenyi post-hoc tests, sensitivity and ablation studies, and centered kernel alignment (CKA) analysis to examine depth-wise feature representation robustness under different distortions. Comprehensive comparisons between the three best performing configurations, namely PRIQA, PDIQA, and PEIQA, and existing state-of-the-art NR-IQA methods demonstrate that the proposed Parasite ResNet-101 IQA (PRIQA) framework achieves superior performance on the constructed parasite image dataset and offers an objective, automated solution for quality control in microscopic parasite imaging. PRIQA can facilitate more consistent downstream detection and diagnosis workflows, minimize manual rescreening, and provide a solid basis for expanding NR-IQA to additional specialized biomedical imaging contexts by consistently detecting low-quality unreliable images. By enabling reliable and automated quality assessment of parasitic microscopy images, this work supports more consistent diagnostic and screening workflows and aligns with the United Nations Sustainable Development Goal 3 (Good Health and Well-Being), particularly in improving disease prevention and diagnostic reliability.

### Research gap

Existing NR-IQA methods have demonstrated efficacy in assessing the quality of natural and medical images. However, their suitability for parasitic image assessment, particularly for microscopic images of Cryptosporidium spp. and Giardia spp., has not been established. Parasitic images present distinct visual challenges, such as low contrast, non-uniform textures, and complex background noise, which differ significantly from natural scene distortions. These differences leave a critical gap in developing reliable quality assessment tools for public health and diagnostic applications. This study aims to address this gap by proposing a robust NR-IQA model specifically tailored to Cryptosporidium spp. and Giardia spp. parasite images.

## Materials and methods

### Training and testing dataset

A total of twenty-three parasite images, each with a resolution of 1376 x 1320 pixels, are obtained from the Department of Parasitology at the University of Malaya, Malaysia, featuring two parasite species: Giardia spp. and

Cryptosporidium spp. The images are captured in the RGB color space as shown in Fig 1 ranging from 0 to 255, ensuring detailed color representation necessary for accurate analysis. These images served as the basis for applying distortion techniques for evaluating both traditional NR-IQA models and the proposed deep convolutional neural network (DCNN)-based models.

The twenty-three reference images were distorted with four types of distortions namely, Gaussian White Noise (GWN), Speckle Noise, Salt and Pepper (SnP) and JPEG Compression at nine levels. Nine levels of distortion were chosen to comprehensively evaluate the model's ability to manage varying degrees of image quality degradation. These distortion types were used as they are the commonly occurred distortion on parasite images. GWN exists in images due to electronic sources, significantly impacts image quality by introducing random variations in pixel intensity, thereby reducing the clarity and accuracy of the captured details [30–32]. Speckle noise is a common occurrence in microscopic images, arising from the interference of scattered light within the sample. This granular pattern can obscure delicate details and reduce image quality, presenting a challenge in accurate cellular imaging [33]. Salt and pepper noise, a prevalent type of impulsive noise, disrupts image quality by introducing random white and black pixels. This interference can obscure details, distort features, and complicate image analysis and segmentation processes [34]. JPEG compression is applied to microscopic images for file storage purposes, and this may introduce artifacts and degradation, potentially affecting segmentation accuracy and subsequent analysis tasks [35,36]. Therefore, it is essential to thoroughly evaluate its impact on image quality, reliability, and the efficacy of automated analysis algorithms. Table 1 shows the types of distortion and its noise levels.

In total, 1,058 images were generated, comprising 207 images for each of the 3 distortions, alongside twenty-three reference images. Fig 2 shows a sample of reference images with its corresponding distorted images.

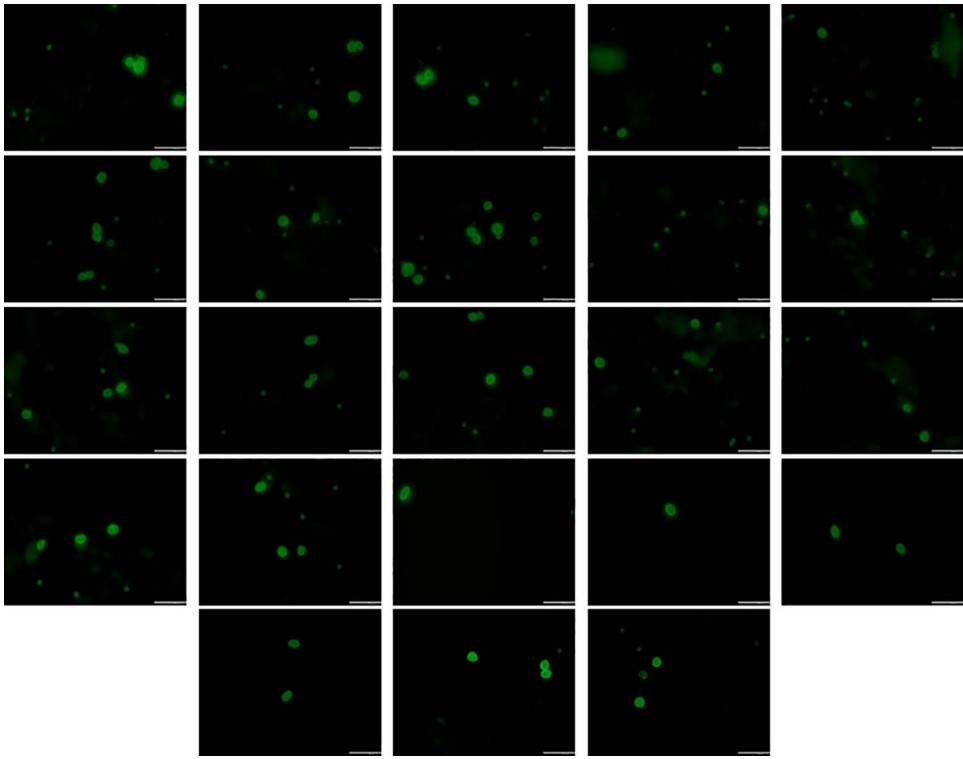

**Fig 1. Twenty-three reference images of Cryptosporidium spp. and Giardia spp. used for dataset generation.**

**Table 1. Distortion types and their noise levels.**

| Type of Distortion | Description | Controlling Parameter |
|---|---|---|
| Gaussian White Noise (GWN) | Standard Deviation (σ): Controls the intensity of the noise. Higher σ results in more intense noise. | $\sigma_{GWN}$: 10, 20, 30, 40, 50, 60, 70, 80 and 90 |
| Salt and Pepper (SnP) | Noise Density (d): Defines the proportion of pixels replaced by noise. Higher d results in more intense noise. | $d_{SnP}$: 0.01, 0.02, 0.03 0.04, 0.05, 0.06, 0.07, 0.08 and 0.09 |
| Speckle Noise | Variance (σ²): The variance controls the intensity of the speckle noise. Higher variance (σ²) results in more intense noise. | $\sigma^2_{Speckle}$: 0.01, 0.02, 0.03 0.04, 0.05, 0.06, 0.07, 0.08 and 0.09 |
| JPEG Compression | Quality Factor (Q): Ranges from 0 to 100. Higher value means less compression and better image quality. Lower value means higher compression and more distortion. | $Q_{JPEG}$: 10, 15, 20, 25, 30, 35, 40, 45 and 50 |

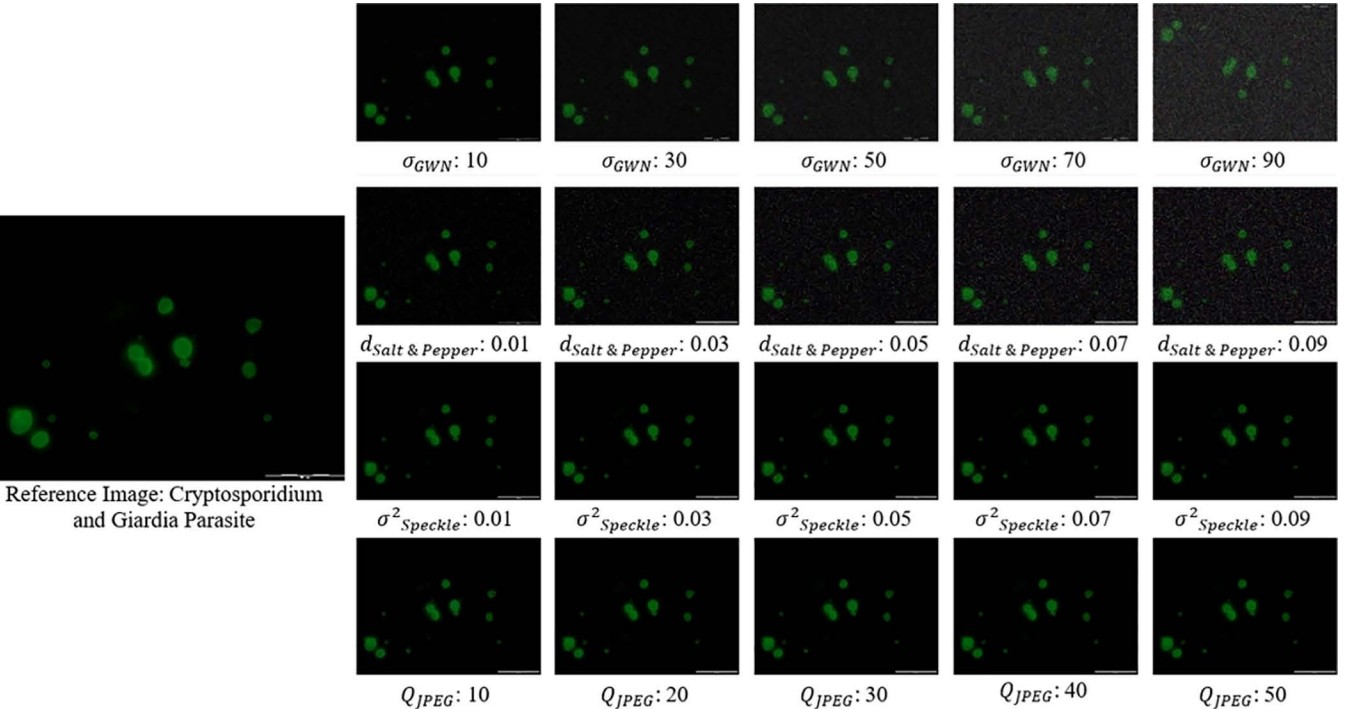

**Fig 2. Sample of a reference parasite image (Cryptosporidium spp. & Giardia spp.) with four different distortion images.**

## Mean Opinion Score (MOS)

Twenty human subjects with normal vision acuity who are aged between 22 and 28 years have been chosen to assess the quality of parasite images. The evaluation followed guidelines outlined in Rec. ITU-R BT.500−11, conducted within an office setting using a 21-inch LED monitor with a resolution of 1920 x 1080 pixels [37]. Before the assessment, each participant's uncorrected near vision acuity was verified using the Snellen Chart to ensure their suitability for the task.

Subjective evaluation employed the Simultaneous Double Stimulus for Continuous Evaluation (SDSCE) methodology, wherein reference and distorted images were presented side-by-side on the monitor screen [38]. The reference image appeared on the left, while its corresponding distorted version was displayed on the right as shown in Fig 3. Participants assessed the quality of the distorted image relative to the reference, assigning ratings of Excellent (5), Good (4), Fair (3),

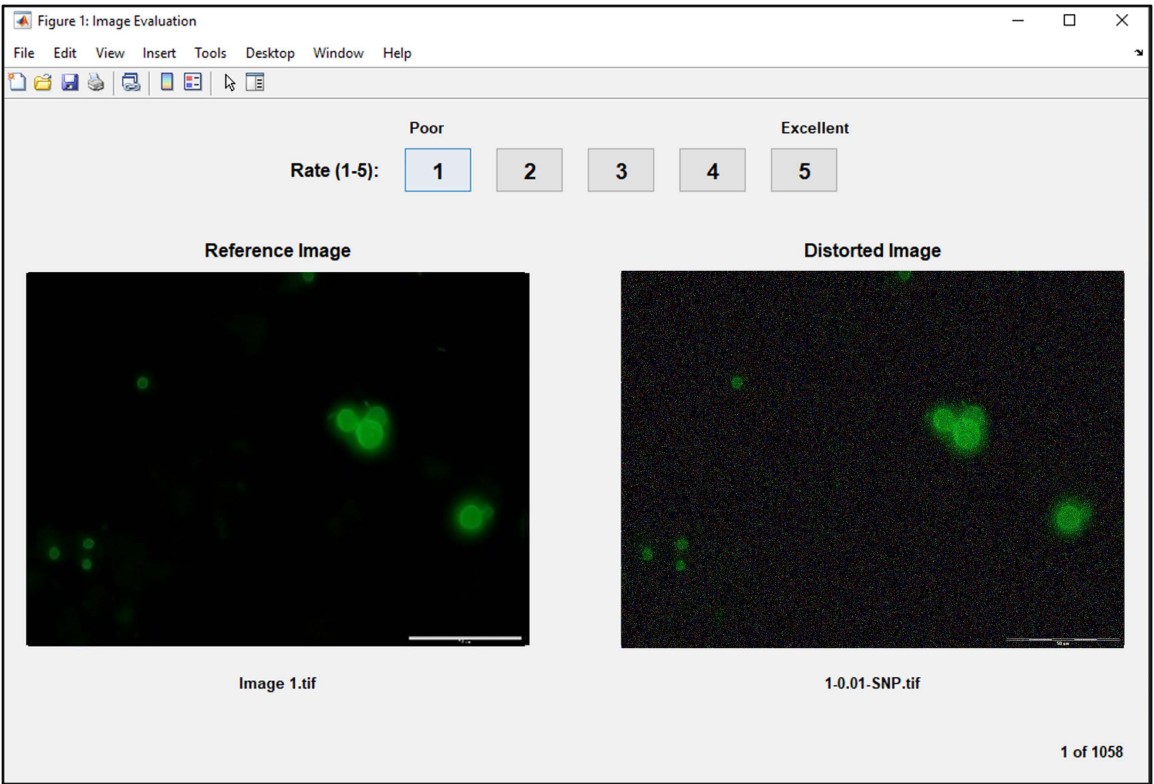

**Fig 3. Sample of subjective evaluation.**

Poor (2), or Bad (1) accordingly. To minimize bias, numerical scores were not disclosed to the participants. Each evaluation session lasted approximately 15–20 minutes.

The ratings provided by the participants were utilized to calculate MOS following established procedures [39]. The ratings obtained from the subjects were used to calculate MOS using Eq. (1) where the average of the scores obtained from all the twenty human subjects were calculated:

$$MOS\,(k) = \frac{1}{N}\sum_{i=1}^{N} S_{ik}$$

(1)

where $S_{ik}$ is the score given by $i^{th}$ subject for $k^{th}$ image and N is the number of human subjects. In this study, N = 20 as we have twenty human subjects.

**Deep Convolutional Neural Network (DCNN)**

The study utilized Deep Convolutional Neural Networks (DCNNs), employing nine distinct models: EfficientNet-B0 [40], DarkNet-53 [41,42], Inception-ResNet-v2 [43], DarkNet-19 [44], ResNet-18 [45], ResNet-50 [45], ResNet-101 [45], GoogLeNet [46], and AlexNet [47]. All backbones were initialized with weights from the MATLAB Deep Learning Toolbox (ImageNet: ~ 1.2 M images, 1000 classes). For inference, each RGB image is resized to the backbone's required input size and normalized with ImageNet mean and standard deviation, then forwarded through the chosen backbone to its late convolutional block. Activations are converted to a fixed-length representation via global average pooling (GAP), yielding

a D-dimensional vector. This vector is mapped to a MOS using a lightweight regression learner trained on the training split; CNN weights remain frozen throughout. We favor GAP layer because it compresses spatial activations into compact, position-agnostic descriptors that preserve global content while reducing overfitting and computation properties well suited to downstream MOS regression. Table 2 summarizes, for each backbone, the input size, selected late layer, feature dimensionality, model size and parameter count, the batch setting used for measurement, and the observed runtime characteristics (latency per image, throughput in FPS) together with peak VRAM.

All experiments were run in MATLAB R2024a with Deep Learning Toolbox and Deep Network Designer on Windows 10 (64-bit) with an NVIDIA GeForce RTX 3060 Ti (8 GB) GPU, an Intel Core i5-10400 CPU, and 16 GB RAM. GPU acceleration was enabled via Parallel Computing Toolbox, inference used FP32 precision. Latency and throughput were profiled on this system. Peak VRAM was recorded with nvidia-smi at steady state. All runs used fixed random seeds for reproducibility.

To assess the robustness and generalization of the models, the study conducted two different cross-validation techniques: 5-fold and 10-fold cross-validation. Utilizing both 5-fold and 10-fold cross-validation provides a comprehensive evaluation of model performance and ensures robustness. Cross-validation helps in assessing how well the model generalizes to an independent dataset. The 5-fold cross-validation is faster in computing and useful for quick checks and when computational resources are limited [48]. In contrast, 10-fold cross-validation provides a more thorough evaluation, often resulting in more reliable and stable estimates of model performance, offering better bias-variance trade-off, which leads to more reliable estimates of performance [49].

Additionally, three different test separation methods were employed, involving partitioning the dataset randomly into 70:30, 80:20, and 90:10 of training and testing respectively. Using different training and testing images allows for evaluating the impact of the size of the test set on the model's performance and ensures robustness of the results across different proportions of training and test data. A 10% test set leaves a larger portion of the data for training, which can be beneficial for the model's learning, especially with smaller datasets where overfitting can be a concern. However, training with a larger dataset will take longer. On the other hand, 20% and 30% test sets provide a more robust and reliable assessment of the model's performance [50–52].

Following feature extraction through CNNs, the study employed various machine learning algorithms to map the features to MOS, ensuring that the proposed NR-IQA could give quality scores similar to human evaluation. The machine learning algorithms used in this study include ten regression models: Linear Regression, Linear Support Vector Machines (SVM), Quadratic SVM, Cubic SVM, Fine Gaussian SVM, Medium Gaussian SVM, Coarse Gaussian SVM, Fine Tree, Medium Tree and Coarse Tree. Table 3 summarizes the purpose and characteristics of each regression model.

**Table 2. Summary of DCNN models, layer usage, features, and runtime/memory characteristics used for feature mapping to MOS.**

| Model | Input (px) | Layer | Feature (dim) | Size (MB) | Parameters (M) | Batch size | Latency (ms) | Throughput (FPS) | Peak VRAM |
|---|---|---|---|---|---|---|---|---|---|
| ResNet-101 | 224-by-224 | pool5 | 2048 | 3.9 | 171 | 64 | 19.1 | 52.3 | 1.41 |
| Darknet-53 | 256-by-256 | avg1 | 1024 | 1.8 | 159 | 64 | 23.0 | 43.5 | 1.34 |
| EfficientNet-B0 | 224-by-224 | efficientnet-b0\|model\|head\| global_average_pooling2d\| GlobAvgPool | 1280 | 2.5 | 20 | 64 | 25.7 | 38.9 | 1.20 |
| DarkNet-19 | 256-by-256 | avg1 | 1000 | 1.8 | 80 | 64 | 16.5 | 60.7 | 1.24 |
| ResNet-18 | 224-by-224 | pool5 | 512 | 0.9 | 45 | 64 | 13.8 | 72.6 | 1.23 |
| Inception-ResNet-v2 | 299-by-299 | avg_pool | 1536 | 3 | 164 | 64 | 37.5 | 26.6 | 1.45 |
| AlexNet | 227-by-227 | pool5 | 9216 | 21 | 233 | 64 | 8.7 | 114.5 | 1.17 |
| ResNet-50 | 224-by-224 | avg_pool | 2048 | 3.6 | 98 | 64 | 16.7 | 59.9 | 1.30 |
| GoogLeNet | 224-by-224 | pool5-7x7_s1 | 1024 | 1.7 | 27 | 64 | 17.9 | 55.9 | 1.23 |

**Table 3. Overview of Regression Models and Their Functions.**

| Regression Model | Description |
| --- | --- |
| Linear Regression | Fits a straight line to the data; models linear relationships between predictors and response. |
| Linear SVM | Maximizes margin between classes (or regression boundary) with linear kernel; useful for linear separability or trends. |
| Quadratic SVM | Includes polynomial terms of degree 2; captures moderate nonlinear relationships. |
| Cubic SVM | Includes polynomial terms of degree 3; captures stronger nonlinear relationships. |
| Fine Gaussian SVM | Uses RBF kernel with small scale; captures very fine-grained, localized nonlinear variations. |
| Medium Gaussian SVM | RBF kernel with moderate scale; balances flexibility and generalization. |
| Coarse Gaussian SVM | RBF kernel with large scale; captures broad, smooth nonlinear trends with reduced risk of overfitting. |
| Fine Tree | Decision tree with splits; models highly detailed, fine distinctions in feature space. |
| Medium Tree | Decision tree with moderate number of splits; balances complexity and interpretability. |
| Coarse Tree | Decision tree with few splits; creates simple, broad decision boundaries to avoid overfitting. |

These regression models were selected to capture a wide range of relationships between the deep CNN features and the subjective MOS scores, from simple linear trends to complex nonlinear patterns. Linear and polynomial SVM models enable testing of different degrees of nonlinearity, while Gaussian SVM variants provide flexibility in modeling localized variations through the choice of kernel scale. Decision trees with varying depth were included to assess performance trade-offs between fine-grained modeling and overfitting control.

Predicted scores were then compared against the ground truth MOS obtained through subjective evaluation using performance metrics. Performance metrics were utilized to evaluate the accuracy and reliability of the predicted image quality score obtained from the best-performing model. In this study, Pearson linear correlation coefficient (PLCC), Root Mean Square Error (RMSE), Spearman rank-order correlation coefficient (SROCC) was used as performance metrics.

Based on the highest values of PLCC, SROCC and the lowest value of RMSE, ResNet-101 model combined with Cubic SVM regression is chosen as the proposed method to predict the image quality score of the parasite images.

The whole experimental workflow is shown in Fig 4 to give a clear understanding of the experimental design, backbone selection, regression learning, and evaluation procedure. This workflow summarizes the steps to identify the most suitable CNN backbone and regression configuration prior to establishing the final framework.

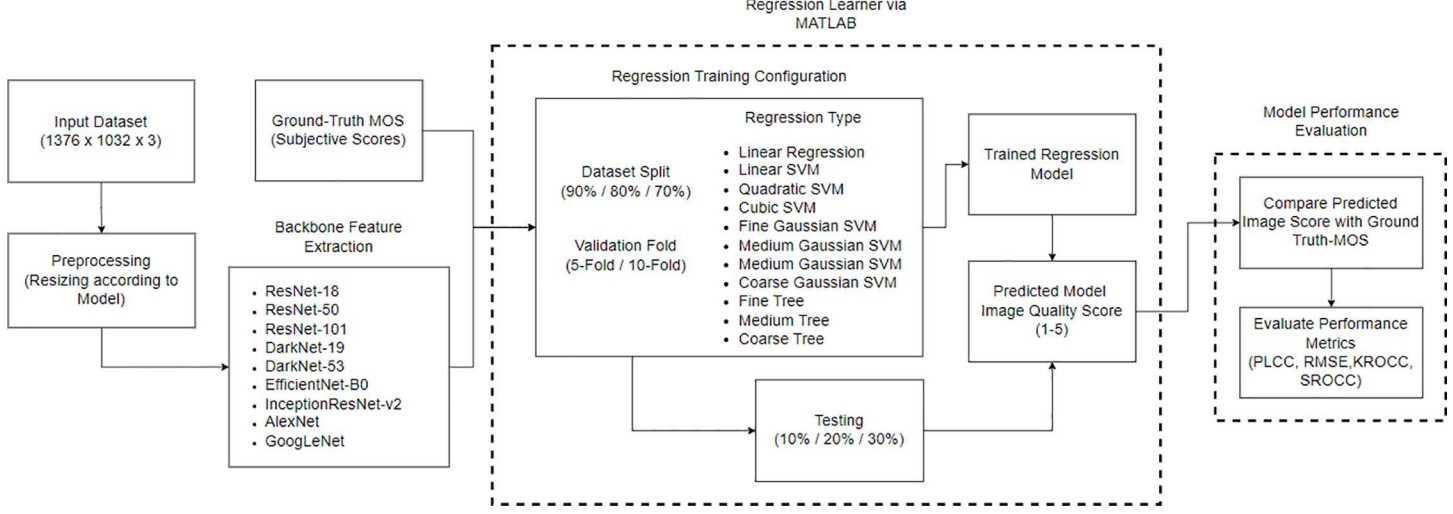

**Fig 4. General framework for CNN-based NR-IQA model selection.**

## Layer-wise CKA analysis

Since distortions can alter feature representation, it is important to measure how closely these representations align under varying conditions. To quantify the impact of distortions on learned representations, we employ linear Centered Kernel Alignment (CKA) [53], a similarity metric that compares centered Gram matrices of feature activations and remains robust across model architectures [54]. For each backbone, we investigate three depths (Early/Mid/Late) as shown in Table 4. For every clean image, we pair it with its distorted counterpart for each distortion (JPEG, GWN, Speckle, SnP) and compute CKA between clean and distorted global pooled features (per-channel spatial average). CKA ∈ [0,1], where higher value indicates greater representational similarity. For compact reporting we present four depth-wise bar charts (one per distortion), showing Early/Mid/Late CKA for each backbone on a common [0,1] scale.

## Proposed framework

The pipeline of proposed Parasite ResNet-101 Image Quality Assessment (PRIQA) framework in Fig 5 starts with input parasite images (1376 x 1032 x 3) undergoing preprocessing steps, including resizing (224 x 224 x 3), to ensure consistent input dimensions for the ResNet-101 backbone. The feature extraction block utilizes convolutional layers to extract hierarchical features of varying resolutions. These features are passed through multiple predictors to generate the required outputs for training and testing the regression models.

The extracted features are mapped to subjective MOS using a regression training block. Once trained, the regression models predict image quality scores during testing. The predicted image quality score, which ranges between 1–5, is the PRIQA score which quantifies the quality of the image. In this scoring scheme, a score of 5 indicates excellent image quality, while a score of 1 reflects extremely poor image quality. Thus, higher scores correspond to better visual quality, as perceived by human evaluators. Performance analysis is conducted by comparing the PRIQA scores with ground truth MOS obtained from subjective evaluations. Metrics such as PLCC, SROCC, and RMSE are used to validate the efficacy of the proposed framework. This modular design ensures that the framework is scalable and adaptable for various image quality assessment tasks, emphasizing its robustness and reliability for parasite image evaluation.

The frameworks for Parasite EfficientNet-B0 Image Quality Assessment (PEIQA) and Parasite DarkNet-53 Image Quality Assessment (PDIQA) follow the same pipeline as PRIQA shown in Fig 4, with the only difference being the CNN backbone used for feature extraction (EfficientNet-B0 and DarkNet-53, respectively). Their extracted features are similarly mapped to MOS using the same set of regression models. The regressor is explained in detail in the Results and Discussions section. This approach allows for a direct and fair comparison between PRIQA, PEIQA, and PDIQA in the subsequent analyses.

**Table 4. Layers Probed for CKA (Early/Mid/Late).**

| CNN Model | Early (layer) | Mid (layer) | Late (layer) |
|---|---|---|---|
| ResNet-101 | conv1 | res4b8_branch2c | res5c_branch2c |
| Darknet-53 | conv1 | conv27 | conv53 |
| EfficientNet-B0 | efficientnet-b0\|model\|stem\|conv2d\|Conv2D | efficientnet-b0\|model\|blocks_8\|conv2d\|Conv2D | efficientnet-b0\|model\|head\|conv2d\|Conv2D |
| DarkNet-19 | conv1 | conv10 | conv19 |
| ResNet-18 | conv1 | res3b_branch2b | res5b_branch2b |
| Inception-ResNet-v2 | conv2d_1 | conv2d_106 | conv7b |
| AlexNet | conv1 | conv3 | conv5 |
| ResNet-50 | conv1 | res4a_branch2c | res5c_branch2c |
| GoogLeNet | conv1-7x7_s2 | inception_4c-3x3_reduce | inception_5b_pool_proj |

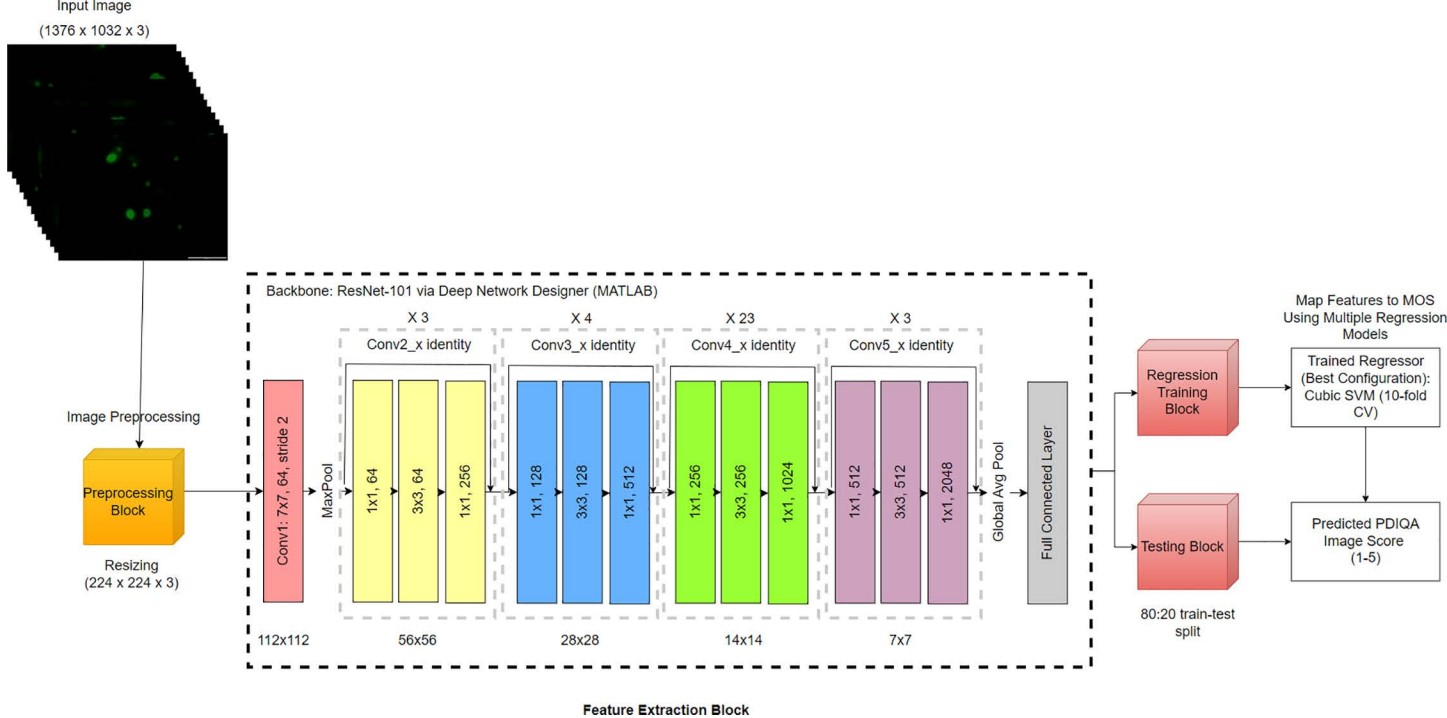

**Fig 5. Pipeline of the proposed Parasite ResNet-101 Image Quality Assessment (PRIQA) framework.**

## No-Reference Image Quality Assessment (NR-IQA)

Following the training and testing phase using DCNNs, the performance of the proposed PRIQA model was further evaluated using another dataset consisting of 125 test images. The PRIQA model was compared with ten state-of-the-art NR-IQA models: EBIQA [55], NIQE [56], BIQI [57]. EPIQA [58], NBIQA [59], ARNIQA [20], ILNIQE [21], BLIINDS II [60], PIQE [61], and BRISQUE [62]. Table 5 shows the development chronology of the ten NR-IQA models along with the model's descriptions.

We include these ten NR-IQA baselines, providing strong and reproducible reference points. They were chosen by four criteria: (i) high citation impact, (ii) public code/pretrained weights with permissive licenses, (iii) reproducibility on commodity CPU/GPU (documented dependencies), and (iv) method diversity (classic NSS vs. modern deep CNN). This criterion ensures transparent replication and complementary inductive biases.

Performance metrics were used to assess the closeness of the quality scores computed by these NR-IQA metrics with the MOS. The predicted scores are said to be close with MOS if the PLCC and SROCC are close to 1, while RMSE values are the least. The closest metric that can predict close to the MOS is the most suitable metric for parasite images.

## Evaluation dataset

The 125 evaluation images were generated using the same distortion types with different levels of distortions to avoid biasness. The predicted quality scores and MOS were obtained for these images and PLCC, SROCC and RMSE were calculated.

## Ethics statement

This study involved human participants solely for the purpose of subjective image quality evaluation. All participants were adults who voluntarily participated, and no personally identifiable or sensitive information was collected. Each participant

**Table 5. Chronological distribution of the development years of the ten selected no-reference image quality assessment (NR-IQA) models.**

| NR-IQA | Year | Description |
|---|---|---|
| BIQI | 2009 | BIQI uses a two-stage framework:<br>1. Classify distortion type using Discrete Cosine Transform (DCT) statistics.<br>2. Predict quality using pre-trained distortion-specific regressors. Mathematically, features $f$ are extracted via DCT subbands, and the output score is:<br>$$BIQI = \sum_{i=1}^{M} p_i \cdot Q_i(f) \qquad (2)$$<br>where $p_i$ is the probability of distortion type $i$, and $Q_i$ is the predicted quality score for that distortion. |
| EBIQA | 2011 | EBIQA evaluates image quality based on the preservation of edges. It extracts edge features using the Sobel operator and compares edge similarity between distorted and reference images:<br>$$EBIQA = \frac{1}{N} \sum_{i=1}^{N} \left| E_r(i) - E_d(i) \right| \qquad (3)$$<br>where $E_r(i)$ and $E_d(i)$ are the edge maps of the reference and distorted images respectively. |
| BLIINDS-II | 2012 | This method works in the DCT domain using NSS features and Bayesian inference:<br>• Divides image into blocks.<br>• Extract features from each block using DCT.<br>• Quality is predicted using a trained Bayesian inference model:<br>$$BLIINDS = E\left[ Q \middle| f \right] \qquad (4)$$<br>where $f$ is the DCT-based feature set. |
| BRISQUE | 2012 | BRISQUE uses locally normalized luminance coefficients:<br>$$\hat{x} = \frac{x - \mu}{\sigma + \epsilon} \qquad (5)$$<br>Then extracts NSS features and fits an SVR to predict quality:<br>$$BRISQUE = SVR\left(\hat{x}\right) \qquad (6)$$ |
| NIQE | 2013 | NIQE is a completely blind IQA method that does not require human labels. It uses Natural Scene Statistics (NSS) to model "pristine" image statistics and computes a Mahalanobis distance:<br>$$NIQE = \sqrt{(x - \mu)^T \Sigma^{-1} (x - \mu)} \qquad (7)$$<br>Where $x$ is the feature vector from the test image, and $\mu$, $\Sigma$ are the mean and covariance of features from natural pristine images. |
| PIQE | 2015 | PIQE uses local variance, block sharpness, and saturation metrics to compute quality. It divides the image into patches and assigns distortion scores to each block:<br>$$PIQE = \frac{1}{N} \sum_{i=1}^{N} d_i \qquad (8)$$<br>where $d_i$ is the distortion score of the $i$-th block. |
| ILNIQE | 2015 | An improvement over NIQE, IL-NIQE integrates features from multiple color channels and orientations:<br>• Extracts NSS features from multiple domains (luminance, chrominance)<br>• Models them with MVG (Multivariate Gaussian)<br>• Computes Mahalanobis distance:<br>$$IL-NIQE = \sqrt{(x - \mu)^T \Sigma^{-1} (x - \mu)} \qquad (9)$$ |
| NBIQA | 2019 | NBIQA combines NSS-based spatial domain features and Support Vector Regression (SVR) to predict quality scores:<br>$$NBIQA = SVR(x) \qquad (10)$$<br>Where $x$ is the handcrafted feature vector extracted from spatial statistics. |
| EPIQA | 2022 | EPIQA uses phase congruency and gradient magnitude maps to assess image quality, emphasizing local structure information:<br>$$EPIQA = f(PhaseCongruency, \; GradientMagnitude) \qquad (11)$$<br>It evaluates local similarities using these two features with structural similarity measures. The exact function $f$ depends on implementation but is typically a weighted $d$ pooling. |
| ARNIQA | 2023 | ARNIQA is a self-supervised deep learning model based on SimCLR. It learns a distortion manifold using contrastive learning and maps it to quality using:<br>• A pre-trained ResNet-50 encoder<br>• An MLP projection head No hand-crafted formulas; model structure is:<br>$$ARNIQA = MLP\left(ResNet50(x)\right) \qquad (12)$$ |

received a written declaration form outlining the study's purpose, procedures, potential risks, and their rights, including the option to withdraw at any time without penalty. Consent was documented by the participant's signature on the declaration form prior to participation, and signed forms are retained in the study records. The study consisted exclusively of visual inspection tasks, with no direct human interaction or intervention. These subjective evaluation settings were followed similarly to Rajagopal et al. where all participants do not require any declaration form to be signed [18,38,63,64]. Our survey data was completely anonymous, did not involve sensitive information, and is purely for internal method validation.

## Full-Reference Image Quality Assessment (FR-IQA)

To validate the subjective MOS collected in this study, additional benchmarking was performed using established full-reference image quality assessment (FR-IQA) metrics. Specifically, Structural Similarity Index (SSIM), Multiscale SSIM (MSSIM), Feature Similarity Index (FSIM), and Information Weighted SSIM (IWSSIM) were computed for all distorted images relative to their pristine reference images. Previous study has shown that combining subjective and objective IQA techniques is beneficial when assessing microscopy images, including parasite samples [64].

These FR-IQA metrics quantify structural similarity between reference and distorted images, serving as objective predictors of perceived quality. Table 6 summarizes the key characteristics of each FR-IQA metric included in this study. The resulting FR-IQA scores were statistically compared with human MOS values by computing the Pearson Linear Correlation Coefficient (PLCC) and Spearman Rank-Order Correlation Coefficient (SROCC). High correlation values were interpreted as evidence that the subjective MOS aligns with established objective measures, thereby validating the use of MOS as training targets for regression models in the proposed framework.

## Statistical significance testing of model performance

Non-parametric statistical tests were employed to evaluate whether differences in model performance metrics were statistically meaningful. Initially, the Friedman test was applied to assess the null hypothesis that all models performed equivalently across evaluation conditions. Upon rejection of this null hypothesis, pairwise comparisons were conducted using the Nemenyi test to identify which specific model pairs exhibit statistically significant differences in average ranks. The Nemenyi test is well-suited for multiple comparison procedures following Friedman tests and has been widely used for model performance comparisons without assuming normality [65].

Additionally, the Wilcoxon signed-rank test was applied to compare the proposed CNN-based models (PRIQA, PEIQA, PDIQA) against established NR-IQA baselines in a pairwise manner. This test evaluates whether the median of the paired differences between two models differs significantly from zero. A significance level of $\alpha = 0.05$ was used for all tests. Following established practice [66], p-values were interpreted such that $p < 0.05$ indicates rejection of the null hypothesis in favor of the alternative, meaning that the difference in model performance is statistically significant. Conversely, $p \geq 0.05$ was interpreted as insufficient evidence to reject the null hypothesis, indicating statistically no significant difference.

**Table 6. Full-Reference IQA Metrics.**

| IQA Metric | Description |
|---|---|
| Structural Similarity Index Metrics (SSIM) | Captures the loss in the structure of the image |
| Multiscale SSIM (MS-SSIM) | Mean of SSIM that evaluates overall image quality by using a single overall quality. |
| Feature Similarity (FSIM) | A low-level feature-based image quality assessment which used two types of features: Phase Congruency (PC) and Gradient Magnitude (GM) |
| Information Weighted SSIM (IW-SSIM) | Obtained by combining content weighting with MSSIM |

These statistical analyses ensured that observed differences in performance metrics such as PLCC, SROCC, and RMSE were not attributable to random variability, supporting claims of genuine model performance advantages.

## Results and discussions

### The Relationship between MOS and different distortion levels

The relationship between MOS and distortion levels for four types of distortions Gaussian White Noise (GWN), Salt & Pepper (SnP), Speckle, and JPEG compression is illustrated in Fig 6. Higher MOS values indicate better perceived image quality, while higher distortion levels correspond to more severe degradation.

As seen in the scatter plots, a general decreasing trend in MOS is observed with increasing distortion levels, particularly for GWN and Salt & Pepper distortions. This suggests that human evaluators were sensitive to the degradation introduced by these distortion types. In contrast, the MOS for Speckle noise and JPEG compression remained relatively consistent across distortion levels, implying that the visual impact of these distortions was either less distinguishable or not perceived as severe by human evaluators. These trends confirmed by subjective human

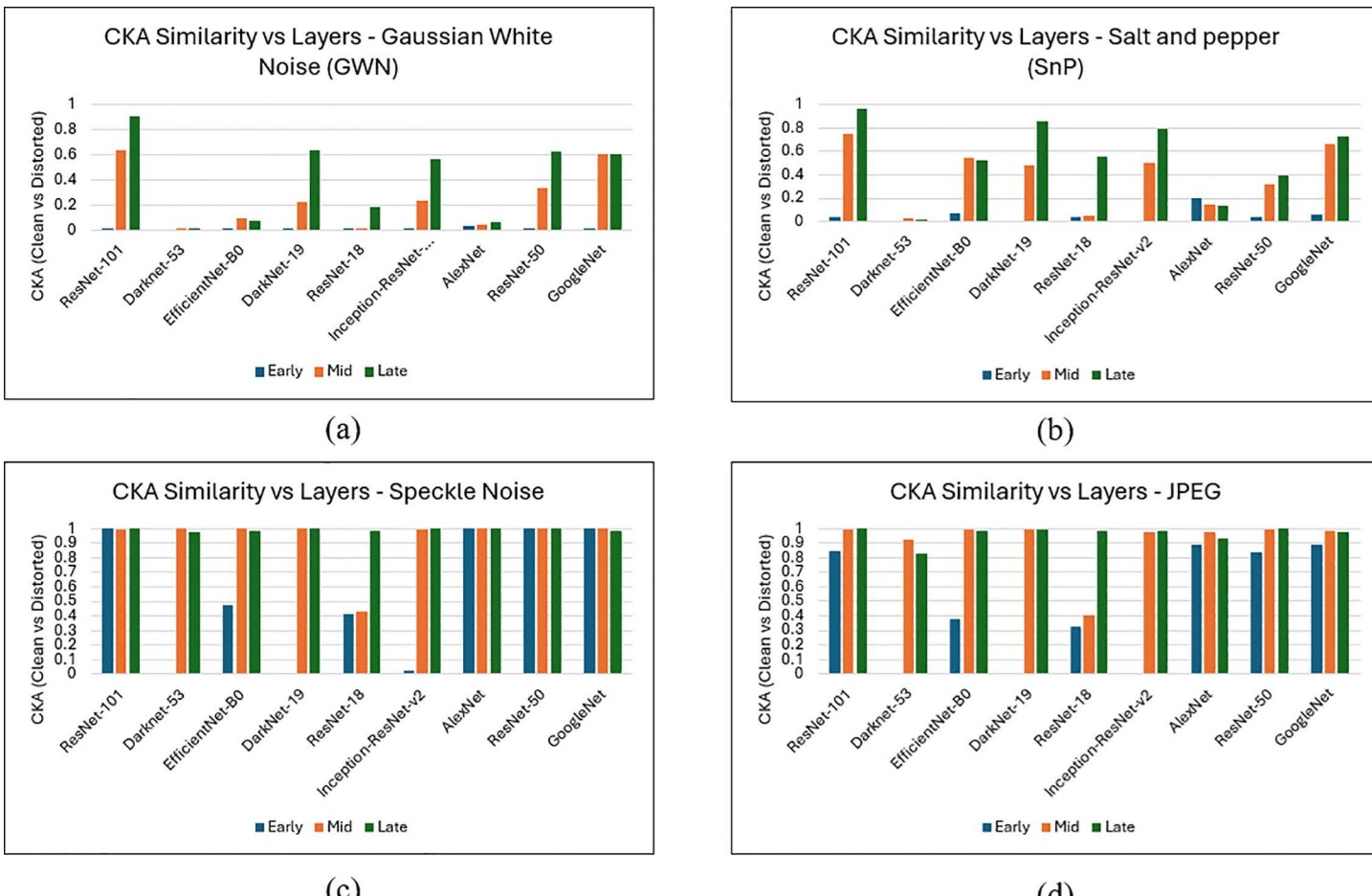

**Fig 6. Scatter plots of Mean Opinion Score (MOS) against distortion levels for each distortion type. (a)** Gaussian White Noise (GWN), **(b)** Salt & Pepper Noise (SnP), **(c)** Speckle Noise, and **(d)** JPEG compression. These plots illustrate the relationship between increasing distortion intensity and subjective image quality as perceived by human evaluators.

evaluation typically correlate inversely with distortion severity except in cases where the artifacts are visually subtle or harder to detect.

## Correlation between MOS and FR-IQA metrics

To assess the validity of the collected MOS values, comparisons were made with FR-IQA scores obtained using SSIM, MSSSIM, FSIM, and IWSSIM. Based on Table 7, the computed PLCC and SROCC values exceeded 0.68 for all four FR-IQA metrics. A study mentioned that a correlation coefficient greater than 0.68 demonstrates that the subjective MOS and FR-IQA scores are closely aligned [38]. Hence, this proved that the results obtained show a strong positive correlation between subjective evaluations and objective structural metrics. This result validates the use of MOS as a reliable target for training regression models and demonstrates that subjective quality judgments of parasite images are consistent with established FR-IQA methods.

   Table 7 shows that the subjective MOS have strong positive correlations with the FR-IQA metrics evaluated. PLCC values were above 0.88 and SROCC values above 0.74 for all metrics, with SSIM and MSSIM showing the highest values. These results suggest that the MOS scores used in this study effectively capture quality differences in a way that aligns with established objective measures. This supports the use of MOS as a reliable ground truth for training regression models. Overall, these findings confirm that the approach can model human-perceived image quality in parasite images, which is important for developing automated inspection tools.

## Performance of the Deep Convolutional Neural Network (DCNN)

Feature extraction was performed using the respective DCNN architectures, and multiple regression methods were subsequently applied to map the extracted features to MOS. The best regression model for each DCNN was identified based on achieving the highest PLCC and SROCC values, together with the lowest RMSE. Table 8 presents the selected DCNN architectures along with their corresponding regression models, cross-validation folds, and dataset splits.

**Table 7. Correlation Between MOS and FR-IQA Metrics Using PLCC and SROCC.**

| FR-IQA | PLCC | SROCC |
|---|---|---|
| MSSIM | 0.968 | 0.769 |
| SSIM | 0.969 | 0.770 |
| FSIM | 0.885 | 0.743 |
| IWSSIM | 0.958 | 0.766 |

**Table 8. Summary of Deep CNN Architectures with Dataset Splits, Cross-Validation, and Best Regression Models.**

| DCNN Architecture | Best Dataset Split | Cross-Validation | Best Regression Model | Prediction Speed (image/s) | Training Time (s) |
|---|---|---|---|---|---|
| ResNet-101 | 80:20 | 10-Fold | Cubic SVM | 291.36 | 90.13 |
| DarkNet-53 | 70:30 | 10-Fold | Cubic SVM | 611.69 | 39.30 |
| EfficientNet-B0 | 80:20 | 10-Fold | Quadratic SVM | 984.42 | 35.35 |
| DarkNet-19 | 80:20 | 10-Fold | Quadratic SVM | 1299.44 | 35.41 |
| ResNet-18 | 80:20 | 5-Fold | Cubic SVM | 2544.03 | 18.91 |
| Inception-ResNet-v2 | 80:20 | 10-Fold | Cubic SVM | 583.90 | 91.62 |
| AlexNet | 90:10 | 5-Fold | Medium Gaussian SVM | 36.94 | 243.31 |
| ResNet-50 | 80:20 | 10-Fold | Cubic SVM | 369.74 | 57.05 |
| GoogLeNet | 80:20 | 5-Fold | Cubic SVM | 1514.10 | 62.19 |

Following the model selection process, regression performance of each DCNN was evaluated using PLCC, SROCC, and RMSE metrics, as shown in Table 9. Speckle noise and JPEG compression recorded lower PLCC and SROCC values, indicating these distortions are harder for the models to assess accurately. The lower correlation for JPEG compression is consistent with previous findings in medical imaging IQA [30], while Speckle noise remains an underexplored challenge.

ResNet-101, EfficientNet-B0, and DarkNet-53 achieved higher PLCC and SROCC values with lower RMSE for Gaussian White, Salt and Pepper, and Speckle noises. This suggests these architectures are better suited for quantifying these types of distortions. Among them, ResNet-101 delivered the best overall performance, with PLCC of 0.993, SROCC of 0.921, and RMSE of 0.188. DarkNet-53 and EfficientNet-B0 also performed strongly, closely following ResNet-101 in predictive accuracy.

In addition to predictive accuracy, Table 9 reports each model's prediction speed and training time. These factors are critical for practical deployment, especially in field laboratories or real-time inspection systems. While ResNet-101 achieved the highest accuracy, its prediction speed was moderate (291.36 image/s) compared to faster models like ResNet-18 (2544.03 image/s) and GoogLeNet (1514.10 image/s).

EfficientNet-B0 and DarkNet-53 offered a balanced trade-off, combining high accuracy with faster prediction speeds and shorter training times. These results suggest that while ResNet-101 is ideal for accuracy-critical tasks, EfficientNet-B0 and DarkNet-53 may be more practical choices for real-time or resource-constrained deployments.

## Analysis of ablation

An ablation study was conducted to investigate the impact of different feature extraction layers within each deep convolutional neural network (DCNN) on the overall performance of the regression models. This analysis aimed to determine the most suitable layer that provides high-quality features for predicting image quality in the PRIQA, PEIQA, and PDIQA models.

For each CNN backbone (ResNet-101, EfficientNet-B0, and DarkNet-53), features were extracted from multiple layers, spanning from early to the deepest layers. These features were subsequently passed to the corresponding regressors, and their performance was evaluated using three standard metrics: PLCC, RMSE, and SROCC on the testing dataset.

Table 9. Regression Performance of DCNN Models Across Distortion Types Using PLCC, SROCC, and RMSE Metrics.

|  |  | EfficientNet-b0 | DarkNet-53 | ResNet-101 | DarkNet-19 | AlexNet | GoogLeNet | Inception-ResNet-v2 | ResNet-18 | ResNet-50 |
|---|---|---|---|---|---|---|---|---|---|---|
| PLCC | GWN | 0.989 | 0.986 | 0.978 | 0.967 | 0.978 | 0.977 | 0.987 | 0.988 | 0.985 |
|  | SnP | 0.974 | 0.979 | 0.979 | 0.951 | 0.934 | 0.940 | 0.954 | 0.975 | 0.977 |
|  | Speckle | 0.939 | 0.934 | 0.865 | 0.944 | 0.786 | 0.897 | 0.933 | 0.941 | 0.872 |
|  | JPEG | 0.888 | 0.674 | 0.866 | 0.821 | 0.659 | 0.527 | 0.872 | 0.695 | 0.534 |
|  | All | 0.992 | 0.992 | **0.993** | 0.987 | 0.980 | 0.981 | 0.990 | 0.991 | 0.990 |
| SROCC | GWN | 0.922 | 0.909 | 0.925 | 0.909 | 0.912 | 0.892 | 0.936 | 0.924 | 0.916 |
|  | SnP | 0.938 | 0.941 | 0.957 | 0.911 | 0.900 | 0.874 | 0.922 | 0.934 | 0.947 |
|  | Speckle | 0.834 | 0.837 | 0.771 | 0.844 | 0.736 | 0.780 | 0.834 | 0.834 | 0.796 |
|  | JPEG | 0.296 | 0.126 | 0.202 | 0.205 | 0.241 | 0.162 | 0.258 | 0.275 | 0.133 |
|  | All | 0.919 | 0.915 | **0.921** | 0.907 | 0.912 | 0.889 | 0.914 | 0.912 | 0.907 |
| RMSE | GWN | 0.183 | 0.200 | 0.250 | 0.307 | 0.250 | 0.257 | 0.191 | 0.187 | 0.206 |
|  | SnP | 0.250 | 0.224 | 0.228 | 0.343 | 0.397 | 0.378 | 0.332 | 0.246 | 0.235 |
|  | Speckle | 0.153 | 0.158 | 0.222 | 0.146 | 0.274 | 0.196 | 0.160 | 0.149 | 0.217 |
|  | JPEG | 0.078 | 0.126 | 0.085 | 0.097 | 0.128 | 0.145 | 0.084 | 0.123 | 0.144 |
|  | All | 0.194 | 0.195 | **0.188** | 0.251 | 0.309 | 0.300 | 0.218 | 0.212 | 0.220 |

The results in Table 10 summarize the ablation study outcomes across the top three CNN backbones, detailing the predictive performance of each extracted layer.

For ResNet-101, the final layer *pool5* achieved the highest PLCC (0.942) and SROCC (0.895), along with the lowest RMSE (0.434), indicating a strong correlation with the subjective MOS. Similarly, for DarkNet-53, the *avg1* layer demonstrated the best overall performance. In the case of EfficientNet-B0, the *global_average_pool* layer outperformed earlier layers, achieving a PLCC of 0.938 and SROCC of 0.882.

These findings suggest that deeper layer capture more abstract and semantically rich features that align more closely with human perception of image quality. Therefore, the deepest and best-performing layer from each of the top three CNNs was selected as the final feature representation for the PRIQA, PEIQA, and PDIQA models.

## Analysis of sensitivity

A sensitivity analysis was conducted to examine the impact of key hyperparameters on the regression performance of the top three selected CNN–regressor pairs: ResNet-101 with Cubic SVM, DarkNet-53 with Cubic SVM, and EfficientNet-B0 with Quadratic SVM. This analysis aimed to assess the robustness and reliability of the PRIQA model outputs under controlled perturbations in support vector machine (SVM) parameters. The three hyperparameters studied were Epsilon, Box Constraint, and Kernel Scale. For each parameter, the baseline value was varied by ±25% and ±50%, and model performance was evaluated using Root Mean Square Error (RMSE) on the testing dataset.

To further validate the selection of hyperparameters, the sensitivity study systematically replaced the default "auto" configuration with manual tuning across the three parameters. It was observed that lowering the box constraint, increasing epsilon, and increasing the kernel scale produced a more rigid model, reducing the risk of overfitting but also potentially limiting flexibility. The results, visualized in Fig 7, revealed that while minor gains in RMSE could be achieved through fine-tuning, the performance was generally consistent around the baseline, indicating robustness. Notably, EfficientNet-B0 demonstrated greater sensitivity to box constraint adjustments, while ResNet-101 and DarkNet-53 exhibited more stable performance under parameter shifts.

These findings support the original decision to use the "auto" setting in the main experiments, as it provided a reliable balance between generalization and precision without the risk of overfitting specific distortions. The adaptive nature of the automatic configuration allowed the model to dynamically determine suitable hyperparameter values for the dataset,

**Table 10. Ablation Study on Extracted Feature Layers Using Testing Dataset Results.**

| CNN | Regressor | Extracted Layer | PLCC | RMSE | SROCC |
|---|---|---|---|---|---|
| **ResNet-101** | **Cubic SVM** | res2c_relu | 0.257 | 6.746 | 0.411 |
| | | res3b3_relu | 0.856 | 6.143 | 0.778 |
| | | res4b22_relu | 0.907 | 0.648 | 0.858 |
| | | re5c_relu | 0.867 | 1.151 | 0.797 |
| | | pool5 | **0.942** | **0.434** | **0.895** |
| **DarkNet-53** | **Cubic SVM** | leakyrelu17 | 0.907 | 0.705 | 0.863 |
| | | leakyrelu36 | 0.917 | 0.635 | 0.880 |
| | | leakyrelu50 | 0.919 | 0.599 | 0.886 |
| | | avg1 | **0.940** | **0.442** | **0.888** |
| **EfficientNet-b0** | **Quadratic SVM** | blocks_4 | 0.906 | 0.688 | 0.891 |
| | | blocks_7 | 0.883 | 0.758 | 0.853 |
| | | blocks_14 | 0.858 | 1.374 | 0.822 |
| | | global_average_pool | **0.938** | **0.449** | **0.882** |

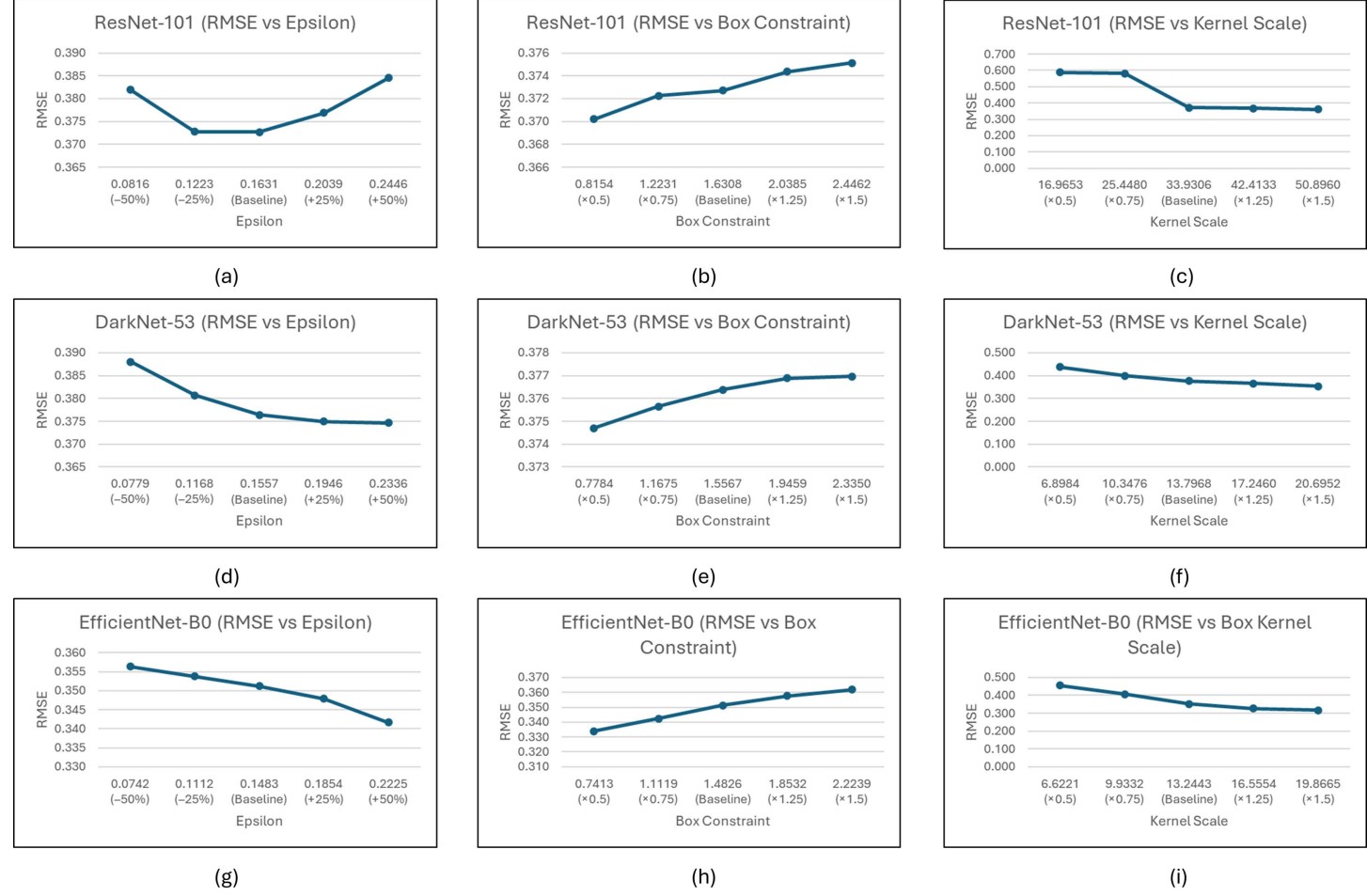

**Fig 7. Sensitivity analysis of RMSE performance for the top-performing models across different SVM hyperparameters. (a–c)** ResNet-101 with Cubic SVM for epsilon, box constraint, and kernel scale, respectively; **(d–f)** DarkNet-53 with Cubic SVM for epsilon, box constraint, and kernel scale, respectively; **(g–i)** EfficientNet-B0 with Quadratic SVM for epsilon, box constraint, and kernel scale, respectively.

which is especially useful in scenarios involving unseen parasite image distributions. Therefore, the default "auto" setting remains a practical and effective strategy for real-world deployment of the PRIQA, PEIQA, and PDIQA frameworks.

### Depth-wise robustness patterns using CKA

The similarity between the learned representations is quantified using Centered Kernel Alignment (CKA). Depth-wise CKA results for GWN, SnP, Speckle and JPEG distortions are illustrated in Fig 8. Based on this analysis, ResNet-101 is selected as the backbone for the proposed PRIQA framework. At the early layer, ResNet-101 is highly sensitive to additive noise at the first layer (GWN = 0.00, SnP = 0.04), while maintaining moderate to high for JPEG = 0.84 and near-identity for Speckle = 1.00. As depth increases in mid layer, substantial recovery is observed by the mid depth (JPEG = 0.99, Speckle = 0.994, SnP = 0.75, GWN = 0.64), indicating progressive invariance with depth. At late depth, ResNet-101 shows near-identity similarity under JPEG and Speckle (0.99–1.00) and strong robustness to additive noise, achieving similarity scores of 0.91 for GWN and 0.97 for SnP. In contrast, several alternatives exhibit weaker late-layer alignment

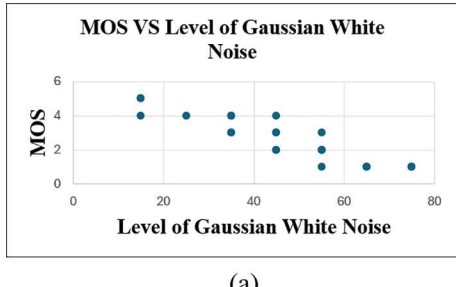

(a)

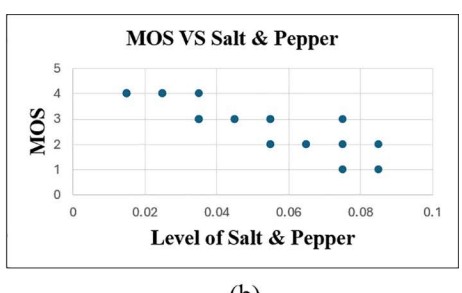

(b)

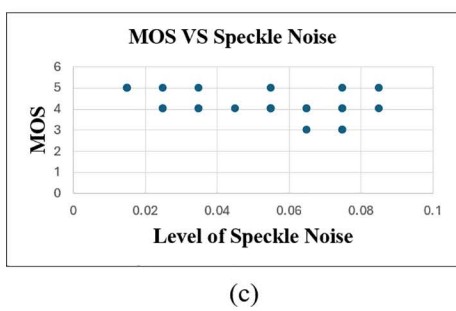

(c)

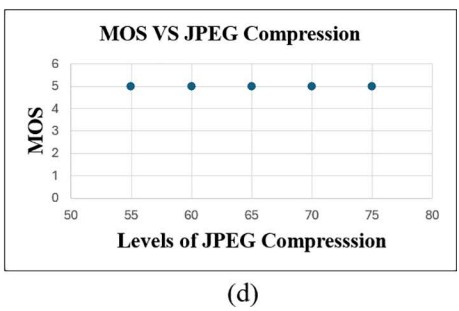

(d)

**Fig 8. Depth-wise linear CKA (clean vs. distorted) across nine backbones. Bars denote Early, Mid, and Late layers for each model in four panels: (a) GWN, (b) Salt and Pepper (SnP), (c) Speckle, and (d) JPEG. Values $\in$ [0,1]; higher = more similar.**

under additive noise, including AlexNet (GWN = 0.06), ResNet-50 (GWN = 0.62), and GoogLeNet (GWN = 0.60), with similar reduced robustness under SnP distortions. Given that microscopy images frequently suffer from the sensor and illumination noise, late-layer robustness is critical for reliable quality assessment. Combined with mature tooling and a stable 2048-D feature at res5c_branch2c or pool5, these findings support ResNet-101 as the most reliable extractor for small-sample dataset.

## Computational complexity and deployment implications

To complement the accuracy analysis, Table 2 reports parameters, latency, throughput, and VRAM to clarify the operational cost of each backbone. When these factors are considered together with the accuracy results, ResNet-101 emerges as the most suitable choice, combining robust late-layer representations and the highest overall accuracy with manageable runtime on commodity hardware (RTX 3060 Ti). For scenarios constrained by throughput or memory, EfficientNet-B0 provides a favorable accuracy–efficiency trade-off and is well suited to field labs or embedded workstations.

## Performance of PRIQA and state-of-the-art NR-IQA models

The analysis from ablation, sensitivity and CKA proved that PRIQA outperformed other model configurations. Furthermore, to verify the performance of PRIQA, the model is compared with other 10 state-of-the-art NR-IQA models. Based on the performance metrics presented in Table 11, the three best-performing models, namely PEIQA (Parasite EfficientNet-B0 Image Quality Assessment), PRIQA (Parasite ResNet-101 Image Quality Assessment), and PDIQA (Parasite DarkNet-53 Image Quality Assessment), were selected for further evaluation using the independent evaluation dataset. These models were subsequently compared against ten state-of-the-art NR-IQA models using the same performance metrics. This comparison aimed to identify the most suitable model for assessing the quality of parasite images, specifically targeting Cryptosporidium spp. and Giardia spp. The comparative performance metrics for PEIQA, PRIQA, PDIQA, and the NR-IQA models are summarized in Table 11. Models that achieve the highest performance metrics are highlighted in

**Table 11. Performance comparison between nR-IQA and proposed DCNN-based models across distortion types.**

| | | BRISQUE | NIQE | PIQE | EBIQA | EPIQA | ILNIQE | ARNIQA | BLIINDS II | NBIQA | BIQI | PEIQA | PDIQA | PRIQA |
|---|---|---|---|---|---|---|---|---|---|---|---|---|---|---|
| PLCC | GWN | 0.492 | 0.855 | 0.685 | 0.931 | 0.860 | 0.412 | 0.700 | 0.736 | 0.894 | 0.702 | 0.959 | 0.956 | 0.949 |
| | SnP | 0.773 | 0.881 | 0.529 | 0.883 | 0.896 | 0.782 | 0.803 | 0.541 | 0.849 | 0.777 | 0.862 | 0.897 | 0.917 |
| | Speckle | 0.088 | 0.002 | 0.197 | 0.460 | 0.481 | 0.115 | 0.102 | 0.203 | 0.016 | 0.157 | 0.738 | 0.610 | 0.730 |
| | JPEG | 0.505 | 0.137 | 0.096 | 0.219 | 0.166 | 0.338 | 0.275 | 0.226 | 0.269 | 0.067 | 0.326 | 0.255 | 0.325 |
| | All | 0.017 | 0.876 | 0.056 | 0.890 | 0.715 | 0.523 | 0.239 | 0.401 | 0.655 | 0.738 | 0.938 | 0.940 | **0.942** |
| SROCC | GWN | 0.001 | 0.873 | 0.894 | 0.931 | 0.893 | 0.410 | 0.699 | 0.812 | 0.885 | 0.543 | 0.954 | 0.925 | 0.944 |
| | SnP | 0.706 | 0.895 | 0.406 | 0.865 | 0.901 | 0.782 | 0.766 | 0.647 | 0.859 | 0.862 | 0.827 | 0.910 | 0.920 |
| | Speckle | 0.037 | 0.075 | 0.230 | 0.320 | 0.418 | 0.116 | 0.080 | 0.385 | 0.156 | 0.131 | 0.685 | 0.606 | 0.656 |
| | JPEG | 0.468 | 0.194 | 0.114 | 0.038 | 0.126 | 0.346 | 0.187 | 0.262 | 0.280 | 0.040 | 0.239 | 0.325 | 0.395 |
| | All | 0.107 | 0.692 | 0.408 | 0.885 | 0.759 | 0.529 | 0.284 | 0.365 | 0.583 | 0.497 | 0.882 | 0.888 | **0.895** |
| RMSE | GWN | 1.158 | 0.690 | 0.969 | 0.487 | 0.679 | 1.212 | 0.951 | 0.901 | 0.597 | 0.947 | 0.377 | 0.392 | 0.421 |
| | SnP | 0.609 | 0.453 | 0.814 | 0.450 | 0.425 | 0.597 | 0.572 | 0.806 | 0.506 | 0.603 | 0.486 | 0.424 | 0.382 |
| | Speckle | 0.630 | 0.633 | 0.620 | 0.564 | 0.554 | 0.628 | 0.632 | 0.619 | 0.632 | 0.625 | 0.427 | 0.501 | 0.432 |
| | JPEG | 0.023 | 0.027 | 0.027 | 0.026 | 0.028 | 0.025 | 0.026 | 0.026 | 0.026 | 0.027 | 0.028 | 0.029 | 0.028 |
| | All | 1.294 | 0.625 | 1.293 | 0.590 | 0.910 | 1.103 | 1.257 | 1.186 | 0.979 | 1.182 | 0.449 | 0.442 | **0.434** |

bold. PRIQA consistently demonstrated superior performance, achieving the highest PLCC, and SROCC values, as well as the lowest RMSE, indicating its efficacy in predicting image quality. The enhanced performance of PRIQA with 10-fold cross-validation is attributed to its ability to leverage a larger training dataset in each fold, thereby facilitating more robust learning and improved generalization.

## Statistical significance test

To evaluate the significance of performance differences between the proposed CNN-based models (PEIQA, PDIQA, and PRIQA) and existing NR-IQA baselines, two complementary non-parametric statistical tests were conducted. First, the Wilcoxon signed-rank test was applied to compare PRIQA specifically against each baseline. This test provides a focused evaluation of whether PRIQA achieves consistent performance improvements over other methods. Table 12 summaries which pairwise comparisons were statistically significant, with significance determined at $p < 0.05$. Second, the Nemenyi

**Table 12. Wilcoxon signed-rank test results comparing PRIQA to other models.**

| Compared Pair | Significance = p < 0.05 |
|---|---|
| PRIQA vs PEIQA | Not Significant |
| PRIQA vs PDIQA | Significant |
| PRIQA vs BRISQUE | Significant |
| PRIQA vs NIQE | Significant |
| PRIQA vs PIQE | Significant |
| PRIQA vs EBIQA | Significant |
| PRIQA vs EPIQA | Significant |
| PRIQA vs ILNIQE | Significant |
| PRIQA vs ARNIQA | Significant |
| PRIQA vs BLIINDS2 | Significant |
| PRIQA vs NBIQA | Significant |
| PRIQA vs BIQI | Significant |

post-hoc test was used for multiple pairwise comparisons among all models to determine whether differences in average ranks were statistically significant. This test offers a broader comparison across all tested models. Table 13 presents the pairwise significance matrix from the Nemenyi test, where a value of '1' indicates a significant difference in ranking between two models, and '0' indicates no significant difference.

The Wilcoxon signed-rank test results in Table 12 show that PRIQA achieved statistically significant improvements over all traditional NR-IQA baselines, reinforcing its superiority for parasite image quality assessment. The only exception was its comparison with PEIQA, which showed no significant difference, suggesting these two CNN-based approaches deliver statistically comparable performance. The results of the Nemenyi test in Table 13 further support these findings by showing that the proposed CNN-based models generally achieved significantly different and better ranks compared to traditional NR-IQA baselines, as indicated by the numerous '1's in their comparisons with existing methods. In contrast, comparisons among PEIQA, PDIQA, and PRIQA themselves show frequent '0's, suggesting no statistically significant difference in ranks among these proposed models. This indicates they perform similarly well in ranking-based evaluations.

## Conclusion

This study proposes the first dedicated NR-IQA framework developed explicitly for parasitic microscopy images, with particular emphasis on Cryptosporidium spp. and Giardia spp., which pose significant public health risks due to their association with waterborne disease outbreaks. The proposed PRIQA framework, based on a ResNet-101 backbone and trained using MOS, demonstrated strong performance across standard IQA metrics (PLCC, SROCC, and RMSE), particularly under 10-fold cross-validation. Compared with PEIQA, PDIQA, and ten state-of-the-art NR-IQA methods, PRIQA consistently achieved higher accuracy in predicting distortion severity without requiring pristine reference images. This enables automated identification of low-quality microscopy images prior to downstream analysis, reducing manual rescreening, mitigating missed detections caused by poor image quality, and supporting more consistent diagnostic decisions across laboratories.

Layer-wise CKA analysis further supports the selection of ResNet-101, showing high feature stability at deeper layers under common distortions (e.g., JPEG, speckle, and additive noise), which facilitates reliable MOS regression while maintaining manageable computational cost for deployment.

**Table 13. Pairwise statistical significance matrix from Nemenyi Post-Hoc Test.**

| Models | PEIQA | PDIQA | PRIQA | BRISQUE | NIQE | PIQE | EBIQA | EPIQA | ILNIQE | ARNIQA | BLIINDS II | NBIQA | BIQI |
|---|---|---|---|---|---|---|---|---|---|---|---|---|---|
| PEIQA | – | 0 | 0 | 1 | 1 | 1 | 1 | 1 | 1 | 1 | 1 | 1 | 1 |
| PDIQA | 0 | – | 0 | 1 | 1 | 1 | 1 | 1 | 1 | 1 | 1 | 1 | 1 |
| PRIQA | 0 | 0 | – | 1 | 1 | 1 | 1 | 1 | 1 | 1 | 1 | 1 | 1 |
| BRISQUE | 1 | 1 | 1 | – | 1 | 0 | 1 | 0 | 1 | 0 | 1 | 0 | 0 |
| NIQE | 1 | 1 | 1 | 1 | – | 1 | 0 | 1 | 1 | 0 | 1 | 0 | 1 |
| PIQE | 1 | 1 | 1 | 0 | 1 | – | 1 | 0 | 0 | 1 | 0 | 1 | 0 |
| EBIQA | 1 | 1 | 1 | 1 | 0 | 1 | – | 1 | 1 | 0 | 1 | 1 | 1 |
| EPIQA | 1 | 1 | 1 | 0 | 1 | 0 | 1 | – | 1 | 1 | 1 | 0 | 0 |
| ILNIQE | 1 | 1 | 1 | 1 | 1 | 0 | 1 | 1 | – | 1 | 0 | 1 | 1 |
| ARNIQA | 1 | 1 | 1 | 1 | 0 | 1 | 0 | 1 | 1 | – | 1 | 0 | 1 |
| BLIINDS II | 1 | 1 | 1 | 1 | 1 | 0 | 1 | 1 | 0 | 1 | – | 1 | 0 |
| NBIQA | 1 | 1 | 1 | 0 | 0 | 1 | 1 | 0 | 1 | 0 | 1 | – | 1 |
| BIQI | 1 | 1 | 1 | 0 | 1 | 0 | 1 | 0 | 1 | 1 | 0 | 1 | – |

('0'= Not Significant, '1' = Significant,' –'= Not Applicable)

Despite its competitive performance, this work has limitations that motivate future research. The current dataset comprises synthetically distorted images under controlled conditions, and model performance on naturally distorted or clinically acquired images from diverse microscopes remains to be validated. Future studies should incorporate broader real-world datasets, explore end-to-end deep learning architectures for joint feature learning and quality prediction, and investigate lightweight or real-time implementations suitable for resource-constrained or point-of-care settings.

Overall, this work advances microscopic image quality assessment by introducing a parasite-specific NR-IQA framework that supports quality-aware automated microscopy and diagnostic workflows. By facilitating more reliable image-based screening for parasitic infections, the study aligns with the United Nations Sustainable Development Goal 3 (Good Health and Well-Being) and indirectly supports United Nations Sustainable Development Goal 6 (Clean Water and Sanitation) through improved monitoring of waterborne pathogens.

## Author contributions

**Conceptualization:** Muhammad Amirul Aiman Asri, Heshalini Rajagopal, Norrima Mokhtar.

**Data curation:** Muhammad Amirul Aiman Asri, Heshalini Rajagopal, Norrima Mokhtar, Yvonne Ai Lian Lim.

**Formal analysis:** Muhammad Amirul Aiman Asri, Heshalini Rajagopal, Norrima Mokhtar.

**Funding acquisition:** Masahiro Iwahashi, Ryosuke Harakawa.

**Investigation:** Muhammad Amirul Aiman Asri, Heshalini Rajagopal, Norrima Mokhtar.

**Methodology:** Muhammad Amirul Aiman Asri, Heshalini Rajagopal, Norrima Mokhtar.

**Resources:** Yvonne Ai Lian Lim.

**Supervision:** Muhammad Amirul Aiman Asri, Heshalini Rajagopal, Norrima Mokhtar, Masahiro Iwahashi, Ryosuke Harakawa.

**Validation:** Muhammad Amirul Aiman Asri, Heshalini Rajagopal, Norrima Mokhtar, Wan Amirul Wan Mohd Mahiyiddin.

**Visualization:** Muhammad Amirul Aiman Asri, Heshalini Rajagopal, Norrima Mokhtar.

**Writing – original draft:** Muhammad Amirul Aiman Asri, Masahiro Iwahashi, Ryosuke Harakawa.

**Writing – review & editing:** Muhammad Amirul Aiman Asri, Heshalini Rajagopal, Norrima Mokhtar, Wan Amirul Wan Mohd Mahiyiddin, Yvonne Ai Lian Lim, Masahiro Iwahashi, Ryosuke Harakawa, Fatimah Ibrahim, Takao Ito.

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
