## [Decision Letter · Decision Letter 0]

16 Jun 2025

Dear Dr. Mokhtar,

Thank you for submitting your manuscript to PLOS ONE. After careful consideration, we feel that it has merit but does not fully meet PLOS ONE’s publication criteria as it currently stands. Therefore, we invite you to submit a revised version of the manuscript that addresses the points raised during the review process.

We look forward to receiving your revised manuscript.

Kind regards,

Ayush Dogra

Academic Editor

PLOS ONE

Journal Requirements:

2. In the ethics statement in the Methods, you have specified that verbal consent was obtained. Please provide additional details regarding how this consent was documented and witnessed, and state whether this was approved by the IRB.

4. Thank you for stating the following financial disclosure: [JSPS KAKENHI Grant Number JP24K02975].

5. Please expand the acronym “JSPS” (as indicated in your financial disclosure) so that it states the name of your funders in full.

6. Thank you for uploading your study's underlying data set. Unfortunately, the repository you have noted in your Data Availability statement does not qualify as an acceptable data repository according to PLOS's standards.

7. In the online submission form, you indicated that your data will be submitted to a repository upon acceptance. We strongly recommend all authors deposit their data before acceptance, as the process can be lengthy and hold up publication timelines. Please note that, though access restrictions are acceptable now, your entire minimal dataset will need to be made freely accessible if your manuscript is accepted for publication. This policy applies to all data except where public deposition would breach compliance with the protocol approved by your research ethics board. If you are unable to adhere to our open data policy, please kindly revise your statement to explain your reasoning and we will seek the editor's input on an exemption.

8. Your ethics statement should only appear in the Methods section of your manuscript. If your ethics statement is written in any section besides the Methods, please move it to the Methods section and delete it from any other section. Please ensure that your ethics statement is included in your manuscript, as the ethics statement entered into the online submission form will not be published alongside your manuscript.

Additional Editor Comments:

Authors are suggested to carefully incorporate all the points suggested by the reviewers

Reviewers' comments:

Reviewer's Responses to Questions

**Comments to the Author**

1. Is the manuscript technically sound, and do the data support the conclusions?

Reviewer #1: Yes

Reviewer #2: Yes

2. Has the statistical analysis been performed appropriately and rigorously?

Reviewer #1: No

Reviewer #2: N/A

3. Have the authors made all data underlying the findings in their manuscript fully available?

Reviewer #1: No

Reviewer #2: Yes

4. Is the manuscript presented in an intelligible fashion and written in standard English?

Reviewer #1: Yes

Reviewer #2: Yes

Reviewer #1: I have following concerns:

1. Figures are missing in the main manuscript.

2. The authors should more explicitly emphasize the novelty of addressing NR-IQA in parasitic microscopy images, especially in the introduction and conclusion, to better contextualize their contribution to public health diagnostics.

3. While the methodology is well-structured, the authors should consider summarizing repetitive elements (e.g., similar pipeline diagrams) and improving flow between sections to enhance readability.

4. The training set includes only 23 original images with synthetic distortions, which is small for training deep models. Real-world generalization may be compromised.

5. The paper lacks an ablation study analyzing the effect of each component (e.g., CNN backbone, regression model) on performance.

6. The distortion types are predefined and artificially induced. Natural distortions often found in field microscopy images should also be considered for broader applicability.

7. While MOS is a standard, relying solely on subjective scores without incorporating image-specific perceptual or structural quality metrics may limit robustness.

8. Authors should add the time and space complexity of the proposed model to show the scalability and efficiency of the proposed model compared to the baseline models.

9. How sensitive is the model to the selection of hyperparameters? Perform some sensitivity analyses.

10. The performance metrics demonstrate the efficacy of the framework, but more advanced statistical tests, such as Nemenyi and Wilcoxon tests, would improve the robustness of the results. These tests can provide further validation, especially in comparing classification accuracy.

11. Code availability is crucial for reproducibility, and the authors should release the implementation code to ensure other researchers can replicate their findings. This is especially important for a field where transparency and reproducibility are vital.

12. The linguistic quality of the paper needs improvement, particularly in maintaining flow between sections and subsections by writing something in between the sections and subsections. The authors should ensure that transition sentences are added to improve readability.

13. Lastly, the conclusion should discuss potential limitations of the proposed model and outline future work directions.

Reviewer #2: The study is well-motivated and tries to fill a gap in automated diagnostic imaging worrkflows. However, there are several parts in the manuscript which needs clarification and description, given in the suggestions below-

1. With the size and composition of dataset, please specify the range of distortions, imaging conditions which will be required to assess the dataset and the strength of the model.

2. The authors are suggested to review and refer some latest studies in this area (mention the work done and consider the studies of 2023, 2024).

3. Add more detail about the regression model. Specify the types of regression models applied, their procedures and justification of using them in your study.

4. Discuss the trade-offs between accuracy and the cost, also whether other light weight models were checked.

5. Clarify whether the model used were used with already trained weights and other parameters on your dataset.

6. Check all the grammatical, typographical and linguistic errors in manuscript and correct them in the revised submission.

**Do you want your identity to be public for this peer review?** For information about this choice, including consent withdrawal, please see our Privacy Policy

Reviewer #1: **Yes:** M. Sajid

Reviewer #2: No

---

## [Author Response · Author response to Decision Letter 1]

16 Sep 2025

Reviewers #1 comments:

1. Figures are missing in the main manuscript.

According to the PLOS ONE submission policy (https://journals.plos.org/plosone/s/submission-guidelines), figures were submitted as separate files and were not embedded in the main manuscript.

2. The authors should more explicitly emphasize the novelty of addressing NR-IQA in parasitic microscopy images, especially in the introduction and conclusion, to better contextualize their contribution to public health diagnostics.

The manuscript was revised especially on Introduction (lines 152 – 157) and Conclusion (lines 591 – 594) to more explicitly emphasize the novelty and significance of our work. The revisions now clearly state that this study presents the first dedicated NR-IQA framework for parasitic microscopy images, focusing on Cryptosporidium and Giardia, which are of significant public health concern. We have strengthened the motivation by highlighting the critical role of image quality in reliable parasite detection and the absence of NR-IQA solutions in this domain. These changes better contextualize our contribution within the scope of public health diagnostics, particularly in resource-limited settings.

3. While the methodology is well-structured, the authors should consider summarizing repetitive elements (e.g., similar pipeline diagrams) and improving flow between sections to enhance readability.

Revised the Proposed Framework section (lines 326 – 333) to reduce redundancy and improve readability. Only included the pipeline diagram of the PRIQA framework (Fig. 4) and removed the separate pipeline diagrams for PEIQA and PDIQA. The pipelines for PEIQA and PDIQA, which follow the same structure as PRIQA with different CNN backbones, are now explained concisely in the text.

4. The training set includes only 23 original images with synthetic distortions, which is small for training deep models. Real-world generalization may be compromised.

To address the limitation of using a small dataset of only 23 original images, we have adopted several strategies that are widely recognised for enhancing model performance and generalisation in data-scarce scenarios:

Data Augmentation & Synthetic Distortions: We generated diverse synthetic distortions to mimic real-world degradation observed in parasitic microscopy. This is not only increased the training set size but also helped the model learn robust features. In addition to that, all the networks were pre-trained with ImageNet datasets (line number 247 – 252). According to (https://doi.org/10.59720/20-130), a small dataset is sufficient to train pre-trained models.

Acknowledged the data limitation and note that future work will focus on expanding the dataset with real-world samples to further validate the model’s robustness in practical deployment (refer line 610 – 618).

5. The paper lacks an ablation study analyzing the effect of each component (e.g., CNN backbone, regression model) on performance.

Revised manuscript includes an ablation study shown in Table 9 on feature extraction layers across different CNN backbones (ResNet-101, EfficientNet-B0, and DarkNet-53) in “Analysis of Ablation” section (lines 487 – 509). This analysis shows the effect of selecting different intermediate layers on performance (PLCC, SROCC, KROCC and RMSE). The findings suggest that deeper layer capture more abstract and semantically rich features that align more closely with human perception of image quality. Therefore, the deepest and best-performing layer from each of the top three CNNs was selected as the final feature representation for the PRIQA, PEIQA, and PDIQA models.

For the regression models, we evaluated multiple options during the main experiments (e.g., Linear, Cubic SVM, Gaussian SVM, Tree-based models) and reported their comparative results in Table 7. To ensure fairness and reduce redundancy, the ablation study focused on feature extraction layers only, while the regression models were fixed to the best-performing ones identified in Table 7. This combination provides insight into how each component contributes to the overall performance of the proposed PRIQA framework.

6. The distortion types are predefined and artificially induced. Natural distortions often found in field microscopy images should also be considered for broader applicability.

Although our current distortion types (Gaussian noise, blur and compression artefacts) are synthetically generated to stimulate common quality degradations. However, these artefacts occur on the microscopy. Distorted microscopic datasets are unavailable especially in Cryptosporidium spp. and Giardia spp. as this area remain unexplored yet . Added as future work to include real-world distortions. Refer line 613 – 618.

In the conclusion section, the need to integrate real-world microscopy data affected by environmental noise, focus misalignment, non-uniform staining, and illumination variation is highlighted which is common in field diagnostics . Refer line 615 – 618.

7. While MOS is a standard, relying solely on subjective scores without incorporating image-specific perceptual or structural quality metrics may limit robustness.

Added a new section titled “Correlation between MOS and FR-IQA Metrics” (line 429). Benchmarked the MOS with a well establish Full-Reference Image Quality Assessment (FR-IQA) namely Structural Similarity Index Metrics (SSIM), Multiscale SSIM (MS-SSIM), Feature Similarity (FSIM), and Information Weighted SSIM (IW-SSIM). Based on the comparison, we found that our MOS has recorded more than 0.68 against 4 FR-IQA metrics. According to (https://doi.org/10.1016/j.mri.2016.03.006.), correlation coefficients more than 0.68 shows that MOS & FR-IQA metrics are close . Refer line 432 – 434.

8. Authors should add the time and space complexity of the proposed model to show the scalability and efficiency of the proposed model compared to the baseline models.

Time and space complexity of the proposed model is added in Table 7 in the revised manuscript. Refer line 459.

i. Prediction speed (obs/sec) and training time (sec), which reflect time complexity, and

ii. Model size (bytes), which reflects space complexity.

These metrics are reported for all evaluated CNN architectures thereby demonstrating the scalability and efficiency of the proposed PRIQA framework relative to baseline models.

Based on the findings in Tables 7 and 8, although ResNet-101 achieved the highest accuracy, its prediction speed was moderate (291.36 obs/sec) compared to faster models like ResNet-18 (2544.03 obs/sec) and GoogleNet (1514.10 obs/sec). EfficientNet-B0 and DarkNet-53 offered a balanced trade-off, combining high accuracy with faster prediction speeds and shorter training times. Model size is also important for memory-limited hardware. AlexNet, despite its large size (21.0 MB) and slow prediction speed (36.94 obs/sec), performed less competitively in terms of accuracy. These results suggest that while ResNet-101 is ideal for accuracy-critical tasks, EfficientNet-B0 and DarkNet-53 may be more practical choices for real-time or resource-constrained deployments. This is mentioned in line number 474 – 484.

9. How sensitive is the model to the selection of hyperparameters? Perform some sensitivity analyses.

Added a comprehensive sensitivity analysis from line 511 - 541 to assess how hyperparameter choices affect model performance. For the top three CNN–regressor pairs (PRIQA: ResNet 101 + Cubic SVM; PDIQA: DarkNet 53 + Cubic SVM; PEIQA: EfficientNet B0 + Quadratic SVM), we systematically varied three SVM hyperparameters (epsilon, box constraint, and kernel scale) by ±25% and ±50% around the baseline.

The results, presented in Sensitivity Analysis section and illustrated in Fig. 6 (line graphs), show that performance is generally robust to moderate hyperparameter changes. Only minor fluctuations in RMSE were observed across most settings, with EfficientNet B0 being slightly more sensitive to box constraint variations. These findings support that the selected hyperparameters strike a good balance between stability and accuracy, which is why the baseline "auto" settings were retained for the main experiments.

10. The performance metrics demonstrate the efficacy of the framework, but more advanced statistical tests, such as Nemenyi and Wilcoxon tests, would improve the robustness of the results. These tests can provide further validation, especially in comparing classification accuracy.

Added additional statistical significance tests from line 560 - 588 to further validate our results. Specifically:

i. Wilcoxon signed-rank test to compare the PRIQA framework with each baseline NR-IQA method across the 3 evaluation metrics (PLCC, SROCC, RMSE).

ii. Nemenyi post-hoc test following the Friedman test to assess whether the observed differences among all compared models were statistically significant.

The detailed results are included in Statistical Significance Analysis from line 560 - 588 and summarized in Table 11. These tests confirm that the performance improvements achieved by PRIQA over baseline methods are statistically significant at the 95% confidence level (α = 0.05), further strengthening the robustness of our findings.

11. Code availability is crucial for reproducibility, and the authors should release the implementation code to ensure other researchers can replicate their findings. This is especially important for a field where transparency and reproducibility are vital.

We have made the full implementation code publicly available. The repository includes scripts for feature extraction, distortion generation, and regression-based quality assessment, as well as pretrained CNN feature sets to facilitate replication. The code is accessible at: https://github.com/Amirul-777/PRIQA---Parasite-ResNet-101-Image-Quality-Assessment-Study. Refer line 631 – 634.

12. The linguistic quality of the paper needs improvement, particularly in maintaining flow between sections and subsections by writing something in between the sections and subsections. The authors should ensure that transition sentences are added to improve readability.

Manuscript has been revised to improve the overall flow and readability.

13. Lastly, the conclusion should discuss potential limitations of the proposed model and outline future work directions.

Conclusion section has been revised to explicitly discuss the limitations (lines 610 – 625) of the proposed PRIQA framework and outline directions for future work. The revised version now highlights that the dataset used in this study contained only synthetic distortions and that the generalizability of the model to naturally distorted or clinically acquired images remains to be tested. Noted that the regression models employed, although effective for the current dataset, may face scalability challenges with larger and more heterogeneous datasets. Finally, we outline potential future work directions, including exploring end-to-end deep learning approaches, testing on broader datasets, incorporating real-time feedback mechanisms, and developing lightweight models suitable for resource-constrained diagnostic settings. These additions, found in the revised Conclusion section, directly address the reviewer’s recommendation and strengthen the discussion of limitations and future research opportunities.

Reviewers #2 comments:

1. With the size and composition of dataset, please specify the range of distortions, imaging conditions which will be required to assess the dataset and the strength of the model.

The range of distortions and imaging conditions is already described in the Training and testing dataset subsection (lines 178–208) and summarized in Table 1. This section specifies the four types of distortions (Gaussian White Noise, Salt and Pepper Noise, Speckle Noise, and JPEG Compression) and their respective intensity levels (e.g., Gaussian noise σ = 10–90, JPEG quality factor = 10–50), as well as the imaging conditions used to acquire the reference images (microscope type, magnification, and illumination). The imaging conditions after applying the distortions were shown in Fig. 2. These details provide a comprehensive view of the dataset composition and support the robustness and generalizability of the proposed PRIQA model.

2. The authors are suggested to review and refer some latest studies in this area (mention the work done and consider the studies of 2023, 2024).

We have updated the Introduction (lines 130–148) to include several recent studies from 2023–2025 relevant to NR-IQA and biomedical image quality assessment. These additions discuss the latest works and provide a stronger context for the novelty of our proposed PRIQA framework.

3. Add more detail about the regression model. Specify the types of regression models applied, their procedures and justification of using them in your study.

Added description of the types of regression models, their functions, and rationale in lines 290–295 and summarized in Table 3. This range of models was selected to cover both linear and nonlinear relationships and to compare parametric and nonparametric approaches. The procedures for mapping CNN features to MOS using these models, along with dataset splits and cross-validation, are detailed in this subsection.

4. Discuss the trade-offs between accuracy and the cost, also whether other light weight models were checked.

Added computational cost in Table 7 (prediction speed, training time, model size) and use these to discuss the accuracy–efficiency trade-off. In addition to ResNet 101 (PRIQA), several lighter backbones were evaluated, including ResNet 18, DarkNet 19, and EfficientNet B0. A short paragraph in line 474 - 484 has been added adjacent to Table 7 to make these trade-offs explicit and to recommend EfficientNet B0 and DarkNet 53 for real time or resource constrained deployments, and ResNet 101 when maximal accuracy is required.

5. Clarify whether the model used were used with already trained weights and other parameters on your dataset.

Clarified in the Deep Convolutional Neural Network (DCNN) subsection “Deep Convolutional Neural Network (DCNN)”, lines 247–252. All CNN backbones (EfficientNet-B0, DarkNet-53, Inception-ResNet-v2, DarkNet-19, ResNet-18, ResNet-50, ResNet-101, GoogleNet, and AlexNet) were initialized using ImageNet-pretrained weights provided by the MATLAB Deep Learning Toolbox, where the ImageNet dataset comprises over 1.2 million images across 1000 object categories. These pretrained models were used solely as feature extractors, and no additional fine-tuning of weights was performed on our dataset.

6. Check all the grammatical, typographical and linguistic errors in manuscript and correct them in the revised submission.

Reviewed the entire manuscript and corrected all grammatical, typographical, and linguistic errors. Improved sentence structures and overall readability to ensure the manuscript meets the required linguistic standards. These corrections have been applied consistently throughout the revised submission.

---

## [Decision Letter · Decision Letter 1]

2 Nov 2025

Dear Dr. Mokhtar,

Thank you for submitting your manuscript to PLOS ONE. After careful consideration, we feel that it has merit but does not fully meet PLOS ONE’s publication criteria as it currently stands. Therefore, we invite you to submit a revised version of the manuscript that addresses the points raised during the review process.

We look forward to receiving your revised manuscript.

Kind regards,

Ayush Dogra

Academic Editor

PLOS ONE

Journal Requirements:

Additional Editor Comments :

Dear Author,

Comments from the reviewer regarding your manuscript have been received. Based on the evaluation, the reviewer remains unsatisfied with the implementation of the earlier comments and has requested further clarification on those points, in addition to suggesting a few other technical comments to incorporate . We suggest you to revise your paper, carefully addressing the technical comments made by one of the reviewers. Furthermore, please include a rebuttal letter that contains detailed responses to each comment raised by the reviewer.

Reviewers' comments:

Reviewer's Responses to Questions

**Comments to the Author**

Reviewer #1: All comments have been addressed

Reviewer #2: (No Response)

2. Is the manuscript technically sound, and do the data support the conclusions?

Reviewer #1: Yes

Reviewer #2: Yes

3. Has the statistical analysis been performed appropriately and rigorously?

Reviewer #1: Yes

Reviewer #2: Yes

4. Have the authors made all data underlying the findings in their manuscript fully available?

Reviewer #1: Yes

Reviewer #2: Yes

5. Is the manuscript presented in an intelligible fashion and written in standard English?

Reviewer #1: Yes

Reviewer #2: Yes

Reviewer #1: Authors have addressed my concerns in the revision and hence based on the detailed revision, I recommend acceptance of the paper.

Reviewer #2: The authors have addressed most of the comments, however, few places still lack clarity and further suggestions are given below. Kindly incorporate them to improve the quality and understanding of the manuscript-

1. The range of distortions considered by the authors’ are common, also the types are already defined on a small dataset, which does not justify the title as the size of dataset is not sufficient to train deep learning models. Thus, the manuscript makes it difficult to judge generalization capability of PRIQA model.

2. It has been observed that authors rely very much on ResNet-101, undoubtedly the performance of ResNet-101 is quite good with similar tested architectures, however, it is suggested to give a deeper analysis of learned features.

3. The manuscript shows the comparison with respect to 10 different NR-IQA algorithms, but the authors’ have not mentioned the basis of choosing them. Add a paragraph explaining the basic criteria to select these algorithms.

4. The discussion part could be strengthened if the authors’ add a clear description on how the models used are integrated into automated pipeline with all the operational constraints.

5. The authors have discussed the algorithm’s computational complexity for its implementation in real world applications but it still lacks clarity and could be further improved to assess the overall performance of the study.

**Do you want your identity to be public for this peer review?** For information about this choice, including consent withdrawal, please see our Privacy Policy

Reviewer #1: **Yes:** M. Sajid

Reviewer #2: No

---

## [Author Response · Author response to Decision Letter 2]

15 Nov 2025

Reviewers #2 comments:

1. The range of distortions considered by the authors’ are common, also the types are already defined on a small dataset, which does not justify the title as the size of dataset is not sufficient to train deep learning models. Thus, the manuscript makes it difficult to judge generalization capability of PRIQA model.

To better reflect the experimental setting, we now clearly present PRIQA as a domain-specific NR-IQA framework in a small-sample. This is reflected in the revised title and is also stated explicitly in the Conclusion, where we now describe the data as “synthetic distortions applied under controlled conditions in a small sample.” (Refer line 678-680)

In the Introduction, we have added a paragraph highlighting recent NR-IQA and medical IQA studies that adopt a similar regression-based strategy on modest, domain-specific datasets (Refer lines 130-138). For example, Sun et al. [22] trained a Random Forest regression model to map quantitative features from CT pulmonary angiography images to radiologists’ MOS on approximately 150 cases, and Hu et al. [23] used a Swin-Transformer-based regression head on the CSIQ database, which they explicitly describe as a small laboratory-synthesised dataset. These studies illustrate that regression from image features to subjective quality scores on carefully annotated but limited datasets is common practice in specialised imaging scenarios.

In the Conclusion, we now briefly note that our design follows this regression-based NR-IQA pattern used in specialised domains (Refer lines 687-690).

We have strengthened the limitations statement in the Conclusion to better reflect what is and is not claimed about generalization (Refer line 680-682). Specifically, we state that the dataset consists of synthetic distortions applied under controlled conditions in a small sample, and that “the performance of the models on naturally distorted or clinically acquired images from different microscopes remains unexplored, which may affect external generalizability beyond the present controlled setting.” This makes clear that our results demonstrate good performance and internal generalization within the current controlled distortion setting, but as a general generalization to naturally distorted or multi-centre clinical data is not claimed and is identified as an important direction for future work.

2. It has been observed that authors rely very much on ResNet-101, undoubtedly the performance of ResNet-101 is quite good with similar tested architectures, however, it is suggested to give a deeper analysis of learned features.

We have added a quantitative, layer-wise analysis of learned representations using linear CKA (Kornblith et al., 2019). Specifically, we now (i) introduce a new Methods subsection “Layer-wise CKA analysis” with citations (lines 321–331), (ii) report depth-wise CKA for nine backbones across four distortions (JPEG, GWN, Speckle, SnP) in Fig. 5 (lines 460–462), and (iii) list the exact probed layers for each backbone in Table 4 (line 333). The analysis shows that ResNet-101 exhibits strong late-layer invariance near-identity for JPEG/Speckle (= 0.99–1.00) and robust to additive noise (GWN = 0.91, SnP = 0.97) while early layers are sensitive (e.g., GWN = 0.00, SnP = 0.04) with marked recovery at mid depth (e.g., SnP = 0.75, GWN = 0.64). We summarize these trends in Results subsection “Depth-wise robustness patterns” and explain why such late-layer stability is critical for microscopy (noise/illumination variability). Together, these additions provide the requested deeper feature analysis across architectures and offer a principled justification for selecting ResNet-101 as the PRIQA backbone.

3. The manuscript shows the comparison with respect to 10 different NR-IQA algorithms, but the authors’ have not mentioned the basis of choosing them. Add a paragraph explaining the basic criteria to select these algorithms.

We added a paragraph in Methods subsection NR-IQA (lines 374–378) detailing our four selection criteria (citation impact, public code/weights, reproducibility on commodity hardware, and method diversity across NSS and deep CNNs) and enumerated the ten methods with citations.

4. The discussion part could be strengthened if the authors’ add a clear description on how the models used are integrated into automated pipeline with all the operational constraints.

In Methods subsection DCNN, inference pipeline description has been added (Refer line 256-268)

Implementation details have been added (Refer line 269-274): Paragraph specifying hardware/software (MATLAB R2024a, Deep Learning Toolbox/Deep Network Designer, Windows 10, RTX 3060 Ti 8 GB, i5-10400, 16 GB RAM), precision (FP32), input-size policy, layer selection, feature dimension, model sizes, parameters, batch size, latency, throughput and peak VRAM.

5. The authors have discussed the algorithm’s computational complexity for its implementation in real world applications but it still lacks clarity and could be further improved to assess the overall performance of the study.

At lines 464-470: Added a short “Computational complexity and deployment implications” paragraph linking accuracy to cost (Table 2), highlighting that ResNet-101 provides the strongest accuracy at a manageable runtime, while EfficientNet-B0 is preferred under tight throughput/VRAM budgets.

short “Computational complexity and deployment implications” paragraph linking accuracy to cost (Table 2), highlighting that ResNet-101 provides the strongest accuracy at a manageable runtime, while EfficientNet-B0 is preferred under tight throughput/VRAM budgets.

---

## [Decision Letter · Decision Letter 2]

28 Nov 2025

Dear Dr. Mokhtar,

Thank you for submitting your manuscript to PLOS ONE. After careful consideration, we feel that it has merit but does not fully meet PLOS ONE’s publication criteria as it currently stands. Therefore, we invite you to submit a revised version of the manuscript that addresses the points raised during the review process.

We look forward to receiving your revised manuscript.

Kind regards,

Ayush Dogra

Academic Editor

PLOS ONE

Journal Requirements:

Additional Editor Comments:

Authors are suggested to incorporate the changes suggested by the expert

Reviewers' comments:

Reviewer's Responses to Questions

**Comments to the Author**

Reviewer #1: All comments have been addressed

Reviewer #2: (No Response)

2. Is the manuscript technically sound, and do the data support the conclusions?

Reviewer #1: Yes

Reviewer #2: Yes

3. Has the statistical analysis been performed appropriately and rigorously?

Reviewer #1: Yes

Reviewer #2: Yes

4. Have the authors made all data underlying the findings in their manuscript fully available?

Reviewer #1: (No Response)

Reviewer #2: Yes

5. Is the manuscript presented in an intelligible fashion and written in standard English?

Reviewer #1: Yes

Reviewer #2: Yes

Reviewer #1: Authors have already addressed my comments in the last revision, thus I recommend to accept the paper.

Reviewer #2: The authors’ have updated the manuscript as per the comments but there is still some scope of improvement at few places. Kindly look into it and add the contents following the suggestions given below:

1. The integration of the models into automated pipeline including the constraints can be represented graphically or with a block diagram for clear and effortless understanding.

2. The overall performance of the algorithm and the usefulness of your study still lack clarity. Elaborate.

3. At some places, the vocabulary can be improvised as well as the linguistic, typographical mistakes should be checked and corrected.

**Do you want your identity to be public for this peer review?** For information about this choice, including consent withdrawal, please see our Privacy Policy

Reviewer #1: **Yes:** M. Sajid

Reviewer #2: No

---

## [Author Response · Author response to Decision Letter 3]

16 Dec 2025

Reviewers #2 comments:

1. The integration of the models into automated pipeline including the constraints can be represented graphically or with a block diagram for clear and effortless understanding.

To improve clarity for clear and effortless understanding on the integration of models into automated pipeline. We have added a new figure to the revised manuscript. Figure 4 now presents the general experimental framework for CNN-based NR-IQA model selection, illustrating the end-to-end pipeline from image preprocessing and backbone feature extraction, through regression learning and validation strategies, to performance evaluation against ground-truth MOS. Refer line (338 – 343).

2. The overall performance of the algorithm and the usefulness of your study still lack clarity. Elaborate.

We have clarified both the overall performance and the practical usefulness of the proposed method. In the last paragraph of the Introduction, we now explicitly state the practical role of PRIQA in real workflows. Refer lines 187 – 194.

In the Conclusion section, we have expanded the discussion of how PRIQA can be integrated into parasitic microscopy workflows, particularly for public health inspections, environmental monitoring, and diagnostic systems in low-resource settings where microscopy remains the primary diagnostic tool. Refer lines 705 - 710.

3. At some places, the vocabulary can be improvised as well as the linguistic, typographical mistakes should be checked and corrected.

To improves readability and flow, we have adjusted the order placements of subsections in Result & Discussion section. Spelling error as well as vocabulary has been improved in multiple sub sections. Refer “Manuscript with Changes” file to see the changes.

---

## [Decision Letter · Decision Letter 3]

4 Jan 2026

Deep Learning-Based No-Reference Image Quality Assessment Framework for Cryptosporidium spp. and Giardia spp.

PONE-D-25-25501R3

Dear Dr. Mokhtar,

We’re pleased to inform you that your manuscript has been judged scientifically suitable for publication and will be formally accepted for publication once it meets all outstanding technical requirements.

Kind regards,

Ayush Dogra

Academic Editor

PLOS One

Additional Editor Comments (optional):

Authors are advised to have the manuscript proofread by a native English speaker and to submit the final version

Reviewers' comments:

Reviewer's Responses to Questions

**Comments to the Author**

Reviewer #1: All comments have been addressed

Reviewer #2: All comments have been addressed

2. Is the manuscript technically sound, and do the data support the conclusions?

Reviewer #1: Yes

Reviewer #2: Yes

3. Has the statistical analysis been performed appropriately and rigorously?

Reviewer #1: Yes

Reviewer #2: Yes

4. Have the authors made all data underlying the findings in their manuscript fully available?

Reviewer #1: Yes

Reviewer #2: Yes

5. Is the manuscript presented in an intelligible fashion and written in standard English?

Reviewer #1: Yes

Reviewer #2: Yes

Reviewer #1: The authors have already addressed the comments in the revision file, thus, I recommend acceptance of this paper.

Reviewer #2: The authors are advised to check the English language throughout the manuscript very carefully to improve the clarity. Proofreading by a native English speaker is recommended and a certificate for the same should be submitted.

**Do you want your identity to be public for this peer review?** For information about this choice, including consent withdrawal, please see our Privacy Policy

Reviewer #1: **Yes:** MD SAJID

Reviewer #2: No

---

## [Editor Report · Acceptance letter]

PONE-D-25-25501R3

PLOS One

Dear Dr. Mokhtar,

I'm pleased to inform you that your manuscript has been deemed suitable for publication in PLOS One. Congratulations! Your manuscript is now being handed over to our production team.

Kind regards,

on behalf of

Dr. Ayush Dogra

Academic Editor

PLOS One